# GWAS, MWAS and mGWAS provide insights into precision agriculture based on genotype-dependent microbial effects in foxtail millet

Genetic and environmental factors collectively determine plant growth and yield. In the past 20 years, genome-wide association studies (GWAS) have been conducted on crops to decipher genetic loci that contribute to growth and yield, however, plant genotype appears to be insufficient to explain the trait variations. Here, we unravel the associations between genotypic, phenotypic, and rhizoplane microbiota variables of 827 foxtail millet cultivars by an integrated GWAS, microbiome-wide association studies (MWAS) and microbiome genome-wide association studies (mGWAS) method. We identify 257 rhizoplane microbial biomarkers associated with six key agronomic traits and validated the microbial-mediated growth effects on foxtail millet using marker strains isolated from the field. The rhizoplane microbiota composition is mainly driven by variations in plant genes related to immunity, metabolites, hormone signaling and nutrient uptake. Among these, the host immune gene FLS2 and transcription factor bHLH35 are widely associated with the microbial taxa of the rhizoplane. We further uncover a plant genotype-microbiota interaction network that contributes to phenotype plasticity. The microbial-mediated growth effects on foxtail millet are dependent on the host genotype, suggesting that precision microbiome management could be used to engineer high-yielding cultivars in agriculture systems.

Genome-wide association studies (GWAS) are widely used to dissect the genetic basis of phenotypic variation[1]. In contrast, most studies on plants are focused on how individual genetic loci contribute to trait variation across populations. Over the past decade, several high-quality genotype-phenotype associations have been identified in important crops, including maize[2], rice[3], sorghum[4], cotton[5] and soybean[6]. However, isolating the genes that control complex yield traits has been difficult, thereby limiting the yield and quality of plants obtained through genetic engineering.

Plants establish relationships with soil-borne microbiota, which can fix nitrogen or protect plants from pathogens or stress[7,8]. The plant genotype can alter the root microbiome, revealing an indirect mechanism for host genes to modulate the phenotypic plasticity of plants that is dependent on environmental exposure[9,10]. For instance, the root microbiota can alleviate the effects of environmental stress, regulate plant development and control defense responses[9]. Identifying the bacterial taxa that promote plant growth and health, and elucidating their interactions with host plants, could transform sustainable agricultural practice. Plant nutrient starvation and plant immunity affect plant-microbe associations and, in turn, the adaptability of plants to the environment[11–15]. Further, root-associated microbes can promote their colonization by inducing the root-

✉e-mail: etwang@cemps.ac.cn; liuhuan@genomics.cn

specific transcription factor MYB72 and excretion of scopoletin[16]. Microbiome genome-wide association studies (mGWAS) are used to elucidate the interaction of host genetic variation with the microbiome in human gut[17] and *Arabidopsis thaliana*[18]. Nevertheless, the identity and relative importance of factors that shape host-microbe interactions and the resulting phenotypes remain poorly understood. Moreover, the beneficial effects of bacterial strains on their host are often cultivar- and species-specific, posing a challenge to their general application[19,20].

Foxtail millet (*Setaria italica*) is an important regional crop in arid and semi-arid areas, especially in East Asia, because of its highly efficient water-use and drought tolerance[21]. Foxtail millet also provides an excellent model for studying abiotic stress as most of the foxtail millet accessions are primarily abiotic stress tolerant particularly to drought and salinity. Therefore, unraveling the genetic variability for agronomic traits can broaden the gene pool for marker-aided breeding programs as well as enhance the efficacy of genetic engineering for abiotic stress tolerance[22]. In addition, the yield component traits like main stem panicle weight (MSPW) and per plant grain weight (PGW) in foxtail millet exhibit highly significant positive correlations with the growth indices such as plant height, leaf width, panicle diameter, and main stem diameter[22], indicating the importance of growth promotion for yield increment. Virtually all yield component traits and most agronomic traits of foxtail millet are quantitatively inherited[23]. Although GWAS has revealed key loci for early and late flowering times and blast-resistance in foxtail millet[24,25], the loci associated with plant growth or yield are still not known. Through large-scale sampling and analysis, microbial composition of the root zone microbiota of foxtail millet and its correlation with the yield trait were identified in our previous study. The findings provided insights into potential agricultural improvement of foxtail millet by root microbiota modification[26]. Here, we performed GWAS and microbiome-wide association studies (MWAS) on growth and yield traits of foxtail millet and quantified the effects of marker SNPs and marker microbes on these agronomic traits. We identified marker strains associated with agronomic traits and validated their growth-promoting or growth-suppressing effects on foxtail millet. We then performed mGWAS on root microbiota and uncovered the host genetic variations that impact the assembly of the root-associated microbiota. We found that root microbes affect agronomic phenotypes in a host genotype-dependent manner. Collectively, this work reveals a reciprocal interplay among host genetic variations, the root-associated microbiota and the agronomic traits of crops.

## Results

### GWAS identifies genetic variations associated with agronomic traits in foxtail millet

A total of 827 foxtail millet cultivars collected from China were sequenced and genotyped using common single-nucleotide polymorphisms (SNPs) based on a ~423 Mb *Setaria italica* cv. Zhanggu reference genome (v.2.3)[27]. In total, 161,562 SNPs were detected after stringent steps of quality control, including population stratification and pedigree filtering, individual- and site-level call-rate filtering, and minor allele frequency (MAF) filtering. The SNPs were evenly distributed along chromosomes and the genetic distance for linkage disequilibrium (LD) decay to its half maximum was 9 kb (Supplementary Fig. 1A, B). Phylogenetic analysis based on the genetic SNPs revealed three main groups in the tested foxtail millet cultivars (Supplementary Fig. 1C).

In addition, we planted these 827 foxtail millet cultivars for a field trial in Yangling, China, and measured their agronomic traits (Supplementary Data 1). Twelve agronomic traits were used for further analysis, including six growth traits and six yield traits. The growth traits were mainly composed of top second leaf length (TSLL), top second leaf width (TSLW), main stem height

(MSH), main stem width (MSW), panicle diameter of the main stem (MSPD) and fringe neck length (FNL) while the yield traits were represented by panicle length of the main stem (MSPL), per plant grain weight (PGW), main stem panicle weight (MSPW), hundred kernel weight (HKW), spikelet number of the main stem (MSSN) and grain number per spike (SGN). Genotype–phenotype analysis showed that all 11 traits were significantly heritable except the trait HKW ($H^2 = 0.006$, $P = 0.15$). Growth traits exhibited higher heritability than yield traits, for example, MSPD showed the highest heritability ($H^2 = 0.46$, the broad sense heritability) while PGW showed the lowest heritability ($H^2 = 0.16$) (Supplementary Fig. 2). GWAS on phenotypes was performed to identify the SNPs associated with the growth and yield traits. In total, 86 significant SNP loci and 91 associations for 10 traits (except MSPW and TSLW) were identified under suggestive $P$-value thresholds ($P < 2.01e−5$), some of which were for multiple traits (Fig. 1, Supplementary Data 2). Among these, 15, 16, 11, 10 and 16 significant SNPs co-located on chromosomes 2, 4, 6,7 and 9, respectively. The candidate genes located around the significant signal were analyzed for known molecular functions (Supplementary Data 3). Firstly, several candidate genes responsible for growth and development regulation were observed such as ATG8C, ERF1B, PRR37 and Cyclin-like F-box. For example, the peak SNP signal *si7:30050703* of MSW, located within the genic region of a homolog of ATG8C (autophagy-related protein 8C, Fig. 1B), which functions in the early development of xylem and phloem tissues[28]. Additionally, SNP *si2:6562955* was associated with MSW and near ERF1 (ethylene-responsive transcription factor) (Fig. 1B and Supplementary Data 3). ERF1 is implicated in cambium proliferation[29], which might influence main stem width. Interestingly, the candidate gene PRR37 near the peak SNP *si2:49328133* of the MSPL (Fig. 1D), suppressed heading and showed shorter panicle length than its mutant in rice[30], which might directly regulate the panicle length of foxtail millet. Besides, the peak SNP *si2:6646016* of PGW was located within the genic region of Cyclin-like F-box (Fig. 1C), which controls many crucial processes such as embryogenesis, hormonal responses, seedling development, floral organogenesis, senescence, and pathogen resistance[31,32].

Secondly, numerous drought stress-responsive (PP2C, ARR12, NPF1.2, NPF4.6, WDR26, Plastocyanin-like protein, CPK2a, PIP5K1) and tolerant genes (APX, DTX12, bHLH3, Thioredoxin fold domain containing protein, SAPK9, $Ca^{2+}$-transporting ATPase, InsP3, E3 ubiquitin-protein ligase, MIEL1) whose expression are frequently upregulated and contribute to drought resistance in drought-stressed seedlings, were found to be associated with the growth and yield traits (Supplementary Data 3). For example, the SNP *si2:49320133* that was associated with MSPD was located within the genetic region of PP2C (phosphatidylinositol-specific phospholipase C) (Fig. 1A, Supplementary Data 3), a stress and ABA-responsive gene that is involved in many physiological responses, including salt, drought and osmotic stress, carbon fixation in C4 plants, and inducible plant responses to pathogen[33,34]. In addition, NPF1.2 (protein NRT1/ PTR FAMILY 1.2) near the peak SNP (*si4:27590764*) of MSPD, which functioned as an ABA importer, is important for the regulation of stomatal aperture in inflorescence stems of *Arabidopsis*[35]. Another candidate gene SAPK9 (serine/threonine-protein kinase SAPK9) near the significant SNP (*si3:44029863*) of PGW improves drought tolerance and grain yield in rice by modulating cellular osmotic potential, stomatal closure and stress-responsive gene expression[36] (Supplementary Data 3).

Thirdly, a great number of plant immune responsive genes and pathogen defense genes were also found to be associated with traits, mainly including RPP13, RGA2, RPS2, LRR-RLKs, EF-Tu SYP22, NOG1, BBE, NB-ARC, and WAK2. Finally, several candidate genes responsible for nutrient uptakes such as iron transporter (IRT1, IRT2*)* and

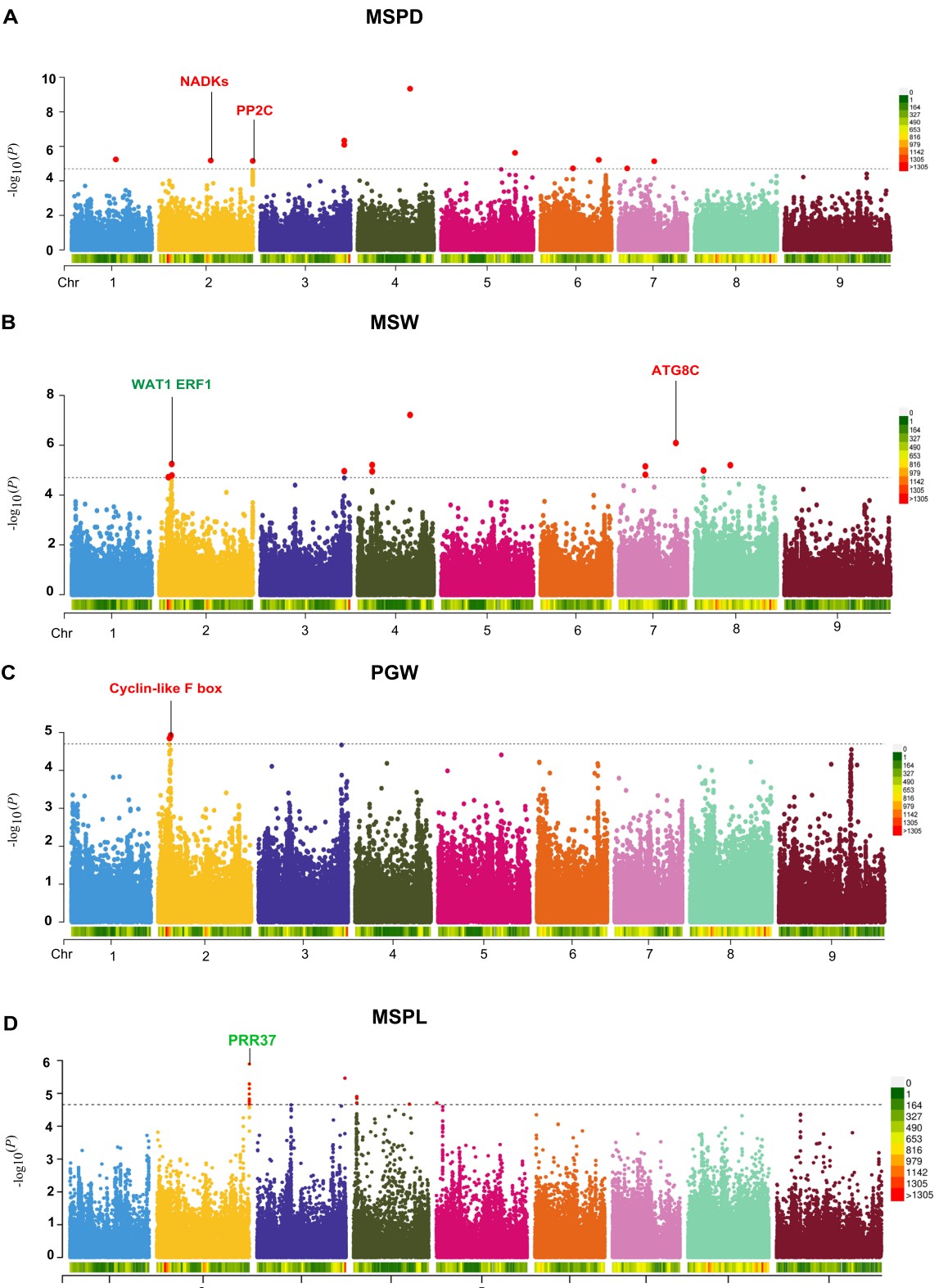

**Fig. 1 | GWAS of agronomic traits in foxtail millet.** Manhattan plots showing the genome-wide associations between host genetic SNPs and **A** panicle diameter of the main stem (MSPD), **B** main stem width (MSW), **C** per plant grain weight (PGW) and **D** panicle length of the main stem (MSPL). The dotted line corresponds to a significance threshold of 2.01e−5. Genes with significant SNPs are marked in red; genes near the significant SNPs are marked in green. NADKs: NAD+ kinase; PP2C:Phosphoinositide phospholipase C 2; WAT1:-WAT1-related protein; ERF1: ethylene-responsive transcription factor 1; ATG8C:autophagy-related protein 8C; PRR37: two-component response regulator-like PRR37.

phosphate transporter (PT) were also observed (Supplementary Data 3). Most of the candidate genes related to the significant SNPs were mainly involved in abiotic and biotic stress responses, implying that the host genotype and environment interaction might co-contribute to plant adaption and modulate the traits of foxtail millet.

## Modeling the effects of genetic and microbial variations on plant phenotypes

To explore the contributions of genetic variations to plant performance, linear regression models were used to calculate the role of host genotypes on key growth (TSLW, MSPD, MSW)- and yield (MSPW, PGW, MSPL)-related traits of the 827 different foxtail millet cultivars. Considering no SNPs associations with phenotypes TSLW and MSPW under suggestive thresholds, we extended the candidate SNPs (adjusted $P < 1.0-e4$) as inputs of the linear regression models[34,35] (Supplementary Data 4). After performing thirty rounds of five-fold cross-validation, the genetic SNP markers in predicting model could explain an average of 32.82%, 28.55%, 47.27%, 15.02%, 38.89% and 64.60% of the variances in TSLW, MSPD, MSW, MSPW, PGW and MSPL in the testing data, respectively (Supplementary Fig. 3).

Root associated microbiota are thought to promote resistance to pathogens and tolerance to specific environmental constraints, and also contribute to plant performance[37,38]. Firstly, linear regression models were performed to calculate the effect of the rhizoplane microbiota on growth- and yield-related traits of the 827 different foxtail millet cultivars. The 1004 rhizoplane operational taxonomic units (OTUs) with a 70% occurrence in all samples (here defined as common OTUs), covering an average of 61.30% of total abundances were used as the input data as these OTUs commonly exist in the root zone of foxtail millet cultivars. The common sub-community (1004 common OTUs) showed higher evenness and correlations with the growth traits than the whole microbial community (Kruskal-Wallis test (one-way analysis), $P_{evenness} = 2.58e-30$, Supplementary Fig. 4A–C). The average variation degree (AVD) index from the common sub-community, 0.5 and 0.3 sub-community (OTUs with 50 and 30% occurrence), were calculated to assess the microbiota stability. The common sub-community had a lower AVD value than the other two sub-communities, indicating that it has a more stable microbiota (Supplementary Fig. 4D). In the common sub-communities, the moderate OTUs (covered 83.67% of OTU numbers) were abundant, followed by abundant OTUs (12.85%) and rare OTUs (3.48%) (Supplementary Table 1). The network analysis was used to disentangle the ecological role and co-occurrence patterns of 1004 OTUs in the common sub-community. Abundant OTUs (ATs) had significantly higher values of the degree, closeness, betweenness centrality and hub scores than both rare (RTs) and moderate OTUs (MTs) in the network (Kruskal-Wallis test (one-way) with $P < 0.001$, Supplementary Fig. 4E), indicating their important roles in sustaining the stability of the microbial community. Thus, the candidate OTUs that were significantly correlated with the traits (adjusted $P < 0.05$) were selected from the common sub-community and used as the input of the predicting models (Supplementary Data 5). A five-fold cross-validation approach was repeated thirty times for each trait to reduce the noise in the estimated model performance. The candidate OTU markers in the OTU-predicting models explained an average of 32.47%, 17.43%, 56.06%, 30.36%, 35.17% and 12.61% of the variances in TSLW, MSPD, MSW, MSPW, PGW and MSPL in the testing data, respectively (Supplementary Fig. 3).

To explore the contributions of genetic variations and environmental microbiota to plant performance, we used linear mixed models to calculate the role of host genotypes and rhizoplane microbiota on the aforementioned growth and yield traits. We used the candidate SNPs (adjusted $P < 1.0-e4$) as inputs of the

linear regression models[39,40] to predict phenotypic variations (Supplementary Data 4). Then the candidate OTUs markers from the above models were also added to the linear regression model (Supplementary Data 5). After performing thirty rounds of five-fold cross-validation, the genetic SNP and OTU markers in predicting model could explain an average of 46.50%, 59.08%, 65.69%, 38.45%, 43.04% and 44.31% of the variances in TSLW, MSPD, MSW, MSPW, PGW and MSPL in the testing data, respectively (Supplementary Fig. 3). The correlation coefficients only using genotype as variables were obviously higher than that only using root microbiota as variables in several agronomic traits such as MSW and MSPL.However, in the trait MSPD and MSPW, the contribution of root microbiota to phenotypic plasticity was higher when root microbiota variables were used instead of genotype variables alone, indicating a different contribution of host genotype and root microbiota to phenotypic plasticity. The combination of host genotype and root microbiota significantly promoted the explanation of variations in all six traits than genotype and root microbiota alone (Wilcox rank test, $P < 0.001$) except for the trait MSPL (Supplementary Fig. 3). Consistently, the panicle length has been proven to be directly impacted by gene PRR37[25], similar to our observed data. The predictive models with the best prediction accuracy for the phenotypes using the SNP and OTU variables were retained, which explained 53.42%, 63.73%, 70.54%, 50.16%, 55.88%, and 54.82% variations for TSLW, MSPD, MSW, MSPW, PGW and MSPL trait, respectively, resulting in a final set of 257 marker OTUs (Fig. 2A–F, Supplementary Data 5). Network analysis of 257 marker OTUs showed that the abundant marker OTUs (AMTs) had a significantly higher value of the degree, closeness and betweenness centrality than both rare (RMTs) and moderate marker OTUs (MMTs), indicating the abundant marker OTUs have more important roles in community structure (Kruskal-Wallis test (one-way) with $P < 0.05$, Supplementary Fig. 5 and Supplementary Table 1).

Among the 257 marker OTUs identified by MWAS, 145 and 128 marker OTUs were significantly correlated with growth and yield traits, respectively (Supplementary Data 5). Taxonomic profiling of these marker OTUs revealed 86 genera distributed across 15 phyla. The top five abundant phyla were Proteobacteria (with 68 OTUs), Actinobacteria (54 OTUs), Bacteroidetes (36 OTUs), Acidobacteria (35 OTUs), and Firmicutes (33 OTUs) (Fig. 3A). In particular,17 marker OTUs were shared by growth and yield traits. Unexpectedly, no marker OTU or genus was shared by all six traits (Supplementary Fig. 6), suggesting that the microbial markers may function in different development stages or different processes of foxtail millet.

## Microbial markers affect foxtail millet growth across different substrates

To validate the predicted effects of these microbial markers on foxtail millet growth, we isolated a range of taxonomically different bacterial strains from root microbiota of the foxtail millet varieties grown in the field. A total of 644 bacterial strains were collected, and 257 bacterial isolates with complete 16 S rRNA gene sequences were retained, representing four bacterial phyla and 25 genera (Supplementary Data 6).

A cultured strain was considered a representative OTU if its 16 S rRNA gene had 97% similarity with the rhizoplane microbiota OTU (Supplementary Data 6). Representative cultivated strains of six positive marker OTUs (Acid550 to *Acidovorax* OTU_46, Baci299 to Bacillaceae OTU_22228, Kita594 to *Kitasatospora* OTU_8, Baci154 to *Bacillus* OTU_19414, Baci312 to *Bacillus* OTU_25704 and Baci429 to Bacillales OTU_381) and four negative marker OTUs (Shin228 to *Shinella* OTU_37, Baci81 to *Bacillus* OTU_54, Baci173 to Bacillaceae OTU_19835 and Baci554 to Bacillaceae OTU_28133) with top beta estimation in the regression model were selected for the validation experiments (Fig. 3A and

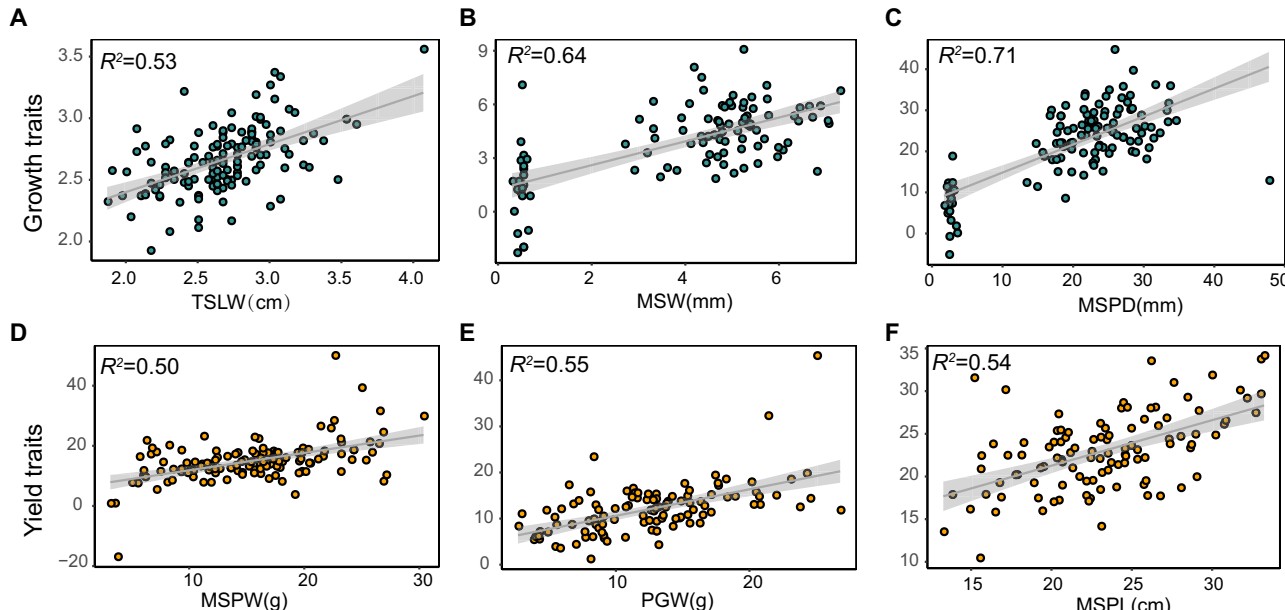

**Fig. 2 | Combined effects of genetic variations and root-associated microbiota on agronomic traits of foxtail millet. A–F** The variation of growth (TSLW, MSW, MSPD) and yield (MSPW, PGW, MSPL) traits explained by the genetic SNPs and microbial OTUs combined. Each panel shows observed values on the x-axis and model-predicted values on the y axis, with a fitted linear regression. Specifically, the predicted value of TSLW, MSW, MSPD, MSPW, PGW, and MSPL is calculated based on 136, 100, 117, 126, 110 and 106 samples in the testing dataset, respectively. The dark trend line illustrates the predicted effect in the linear model (LM). The gray shading around the line represents a confidence interval of 0.95. TSLW, top second leaf width; MSW, main stem width; MSPD, panicle diameter of the main stem; MSPW, main stem panicle weight; PGW, per plant grain weight; MSPL, panicle length of the main stem.

Supplementary Fig. 7). We co-cultivated these 10 biomarker strains with foxtail millet Huagu12 (a bred cultivar of foxtail millet (*Setaria italica*) at Shenzhen, China) for 7-days in sterilized plates, and observed altered root lengths and plant heights compared with the control (Fig. 3B and Supplementary Fig. 7A). The positive biomarker strains representing OTUs with top beta estimation showed significant growth-promoting abilities. Specifically, positive biomarker strain Kita594 (*Kitasatospora* OTU_8) promoted both root and stem growth, whereas Baci299 (*Bacillus* OTU_22228) and Acid550 (*Acidovorax* OTU_46) only promoted shoot growth compared to the control (one-tailed *t*-test with adjusted $P < 0.05$, Fig. 3B and Supplementary Fig. 7A). The negative marker strain Baci173 (Bacillaceae OTU_19835) and Baci554 (Bacillaceae OTU_28133) suppressed the shoot and root growth of Huagu12 (one-tailed *t*-test with adjusted $P < 0.05$, Fig. 3B and Supplementary Fig. 7A). While the negative marker strains Shin228 (*Shinella* OTU 37) and Baci81 (*Bacillus* OTU 54) exhibited growth-promoting effects, they may only function in special root microbial flora in collaboration with other strains or be mistakenly identified as representative strains due to high 16 S rDNA sequence similarities with negative marker OTU 37 and 54.

Next, we validated the effects of four positive marker strains (Kita594, Baci299, Baci154 and Acid550) and two negative marker strains (Baci173 and Baci554) with good promoting or suppressing performances on plant growth in plate experiment by watering millet seedlings grown in sterilized soil with these bacterial suspensions separately. Consistently, the seedlings watered with suspensions of the promoting bacterial strains Kita594, Baci299, Baci154 and Acid550 showed significantly increased plant height and root length compared with the control, whereas the seedlings watered with suspensions of the suppressing bacteria Baci173 showed shorter roots (one-tailed *t*-test with adjusted $P < 0.05$, Fig. 3C and Supplementary Fig. 7B). These results validated the plant growth promoting (PGP) traits of marker microbes in foxtail millet.

## Microbes regulate plant growth via strain-dependent mechanisms

To shed light on how bacterium regulates the growth of foxtail millet, we analyzed the transcriptomes of seedlings colonized for 14 days with the growth-promoting strains Baci299, Acid550, Kita594 or with the growth-suppressing strain Baci173. The differentially expressed genes from biomarker strain-inoculated versus non-inoculated samples were enriched in different pathways (Fisher's exact test, $q < 0.05$, Fig. 4A). For example, the differentially expressed genes caused by growth-promoting strains were mainly enriched in the pathways such as Phenylalanine, tyrosine and tryptophan biosynthesis (ko00400), Biosynthesis of amino acids (ko01230), Phenylalanine metabolism (ko00360), Carbon fixation in photosynthetic organisms (ko00710), Photosynthesis-antenna proteins (ko00196), Photosynthesis (ko00195), MAPK signaling pathway-plant (ko04016), Plant-pathogen interaction (ko04626), Diterpenoid biosynthesis (ko00904), Monoterpenoid biosynthesis (ko00902), alpha−Linolenic acid metabolism (ko00592) and Selenocompound metabolism(ko00450), while the differentially expressed genes caused by suppressing strain were mainly involved in the pathways such as Arginine and proline metabolism (ko00330) and Valine, leucine and isoleucine degradation (ko00280) (Fig. 4A).

Interestingly, the growth-promoting strains displayed strain-specific induction of genes involved in nutrient transformation, pathogen defense, anti-abiotic stresses and growth-promoting processes (Fig. 4B, C, Supplementary Data 7). For instance, the ammonia producing gene (K01455_fomamidase) and terpenoids synthase (K15803, germacrene D) were highly induced by strain 299; ethylene synthase (K05933, aminocyclopropanecarboxylate oxidase) and plant immunity responsive genes (K18834, WRKY1; K20538, MPK8; K00430, peroxidase; K13422, MYC2; and K04079, HSP90A) were abundantly induced by strain 550, and photosynthesis-related genes (K02692, psaD, K01092, IMPA; and K08916, LHCB5), anti-oxidant gene (K00434, L−ascorbate peroxidase) and pterostilbene biosynthesis gene (K16040, ROMT) were highly induced by strain 594

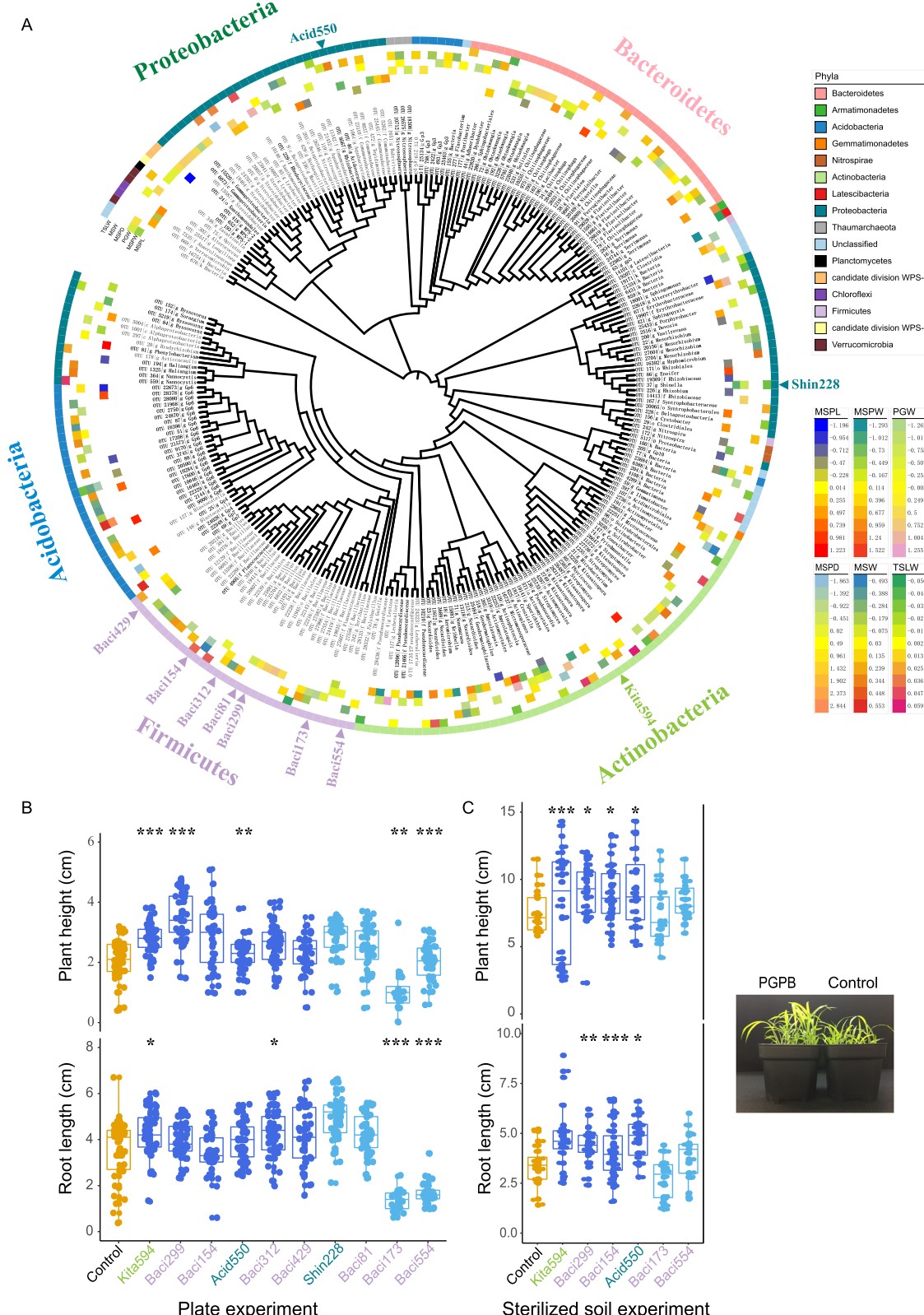

(Fig. 4C). Intriguingly, the expansin gene that mediates cell wall loosening and increased root and shoot growth in rice[41], was induced by all of the growth-promoting strains.

Similarly, 39 genes were significantly induced only by the growth-suppressing strain Baci173, including auxin synthetase (K01426, amidase; K00128, aldehyde dehydrogenases ALDH), auxin-responsive protein IAA (K14484, auxin-responsive protein), L-glutamine synthetase (K01915) and branched-chain amino acid synthetase (BCAT, K00826) (Fig. 4B, C, Supplementary Data 7), which all have well-documented roles in inhibiting root growth[42–44]. Thus, the plant growth mechanisms mediated by microorganisms were strain-dependent.

**Fig. 3 | Microbial markers promote the growth of foxtail millet across different substrates. A** Phylogenetic tree of the 257 microbial markers associated with agronomic traits of foxtail millet. The outer circle represents the phylum level. The beta estimates of the microbial OTUs to growth and yield traits are plotted in the inner circles, respectively. The arrows indicate the strains tested *in planta* (**B**, **C**), including strains responded to six positive marker OTUs: Acid550 to *Acidovorax* OTU_46, Baci299 to Bacillaceae OTU_22228, Kita594 to *Kitasatospora* OTU_8, Baci154 to *Bacillus* OTU_19414, Baci312 to *Bacillus* OTU_25704, Baci429 to *Bacillales* OTU_381, and strains responded to four negative marker OTUs: Shin228 to *Shinella* OTU_37, Baci81 to *Bacillus* OTU_54, Baci173 to Bacillaceae OTU_19835 and Baci554 to Bacillaceae OTU_28133. The strains predicted to affect growth traits are validated by plate (**B**) and sterilized soil (**C**). Significance is determined within each pair of treatment and control via one-tailed *t*-test and the *P*-values are adjusted by

Benjamini-Hochberg (BH) method. $n = 41, 40, 38, 34, 43, 43, 44, 32, 21, 34$ and 27 (from left to right) biological replicates in plate experiment. From Kita594 to Baci554, adjusted $P_{(plant\ height)} = 8.68e{-}07, 1.02e{-}09, 0.07, 0.001, 0.49, 0.29, 0.03, 0.40, 0.001, 5.39e{-}06$, adjusted $P_{(root\ length)} = 0.012, 0.10, 0.40, 0.18, 0.01, 0.11, 1.14e{-}06, 0.02, 1.20e{-}09, 5.68e{-}12$. $n = 20, 32, 31, 40, 23, 20$ and 23 (from left to right) biological replicates in sterilized soil experiment. From Kita594 to Baci554, adjusted $P_{(plant\ height)} = 1.6e{-}07, 0.003, 0.018, 0.011, 0.98, 0.06$, adjusted $P_{(root\ length)} = 0.046, 4.4e{-}04, 8.43e{-}05, 0.016, 0.058, 0.15$. *, ** and *** represented the adjusted $P < 0.05, 0.01$ and $0.001$, respectively. The box depicts the interquartile range (IQR) between the 25th and 75th percentiles, respectively and the line within the box represents the median. The whiskers extend 1.5 times the IQR from the top and bottom of the box, respectively.

## mGWAS-based identification of correlations between host genetic variation and microbiota abundance

To explore the relationship between the host genotype and rhizoplane microbial composition, Mantel's test was used to evaluate the correlation between host phylogenetic distances and rhizoplane microbiota distance, exhibiting a significant Mantel's correlations ($r = 0.06$, $P = 0.0003$, 9999 permutations). Subsequently, to investigate host genotype-dependent variation in the foxtail millet rhizoplane microbiota, the heritable microbes were identified based on a common rhizoplane OTUs data set, which covered 17 phylum and 52 orders. Using an SNP-based approach, the heritability for individual OTU was calculated. 281 OTUs with $H^2$ (the broad sense heritability) more than 0.15 were defined as highly heritable and the others as lowly heritable (Supplementary Data 8). Bacillales and Gp4 orders enriched greater numbers of highly heritable OTUs when compared with the lowly heritable fraction (Fisher's exact test, $q < 0.05$, Supplementary Fig. 8A), implying that these bacterial orders were more easily impacted by host genotypes of foxtail millet. To explore whether there are similarities in heritable microbes across Poaceae family, we compared the top 100 most heritable OTUs from foxtail millet, sorghum[45] and maize datasets[46,47]. After removing the order with a total number of OTUs less than 4, seven bacterial orders such as Bacillales, Actinomycetales, Burkholderiales, Rhizobiales, Myxococcales, Sphingobacteriales and Xanthomonadales were identified, which shared and covered more than half of the most heritable OTUs from foxtail millet, sorghum, and maize datasets, respectively (Supplementary Fig. 8B, C). These results hence indicated that the microorganisms in these bacterial orders were more sensitive to genetic variations across both sorghum, maize and foxtail millet.

To further assess the association of host genetic variations and root microbial abundance, we ran mGWAS on 1004 common rhizoplane OTUs of foxtail millet. We identified significant associations of 2108 SNP loci with 838 microbial OTUs (here called SNPs-associated OTUs) at the genome-wide suggestive significance threshold of $P < 2.01e{-}5$ (Supplementary Data 9). To identify how the host genetic variations drove abundance variations of the specific microbial taxonomies, especially the bacterial orders that were more sensitive to genetic variations, the SNP-associated genes for each order were enriched into pathways (Supplementary Fig. 9). However, only four bacterial orders associated genes were significantly enriched into different pathways. Taking Bacillales for example, the associated genes were mainly enriched in the monoterpenoid biosynthesis pathway (Fisher's exact test, $q = 0.05$, Supplementary Fig. 9). The GP4 associated genes were significantly enriched in producing D-galacturonic acid (Fisher's exact test, $q = 0.08$, Supplementary Fig. 9). GP4 from Acidobacteria phylum, which has been reported with the capability of utilizing galacturonic acid, a characteristic component of the cell wall in higher plant[48], might be recruited to rhizoplane by plant-secreted galacturonic acid. The genes associated with plant pathogen-containing order Xanthomonadales were significantly enriched into the pathway such as peroxisome and MAPK signaling pathway (Fisher's

exact test, $q = 0.01$ and 0.04, Supplementary Fig. 9), which are involved in disease and abiotic resistance[49,50]. These results provide key insights into how the host genetic mechanism drive plant-associated microbiota.

In addition, significant SNP loci located in the generic region were also deeply analyzed (Fig. 5, Supplementary Data 9). For example, the peak SNP signal *si7:13687399* located in the genic region of bHLH35 was associated with 39 common OTUs from different microbial taxonomies such as Acidobacteria (28), Proteobacteria (8) and Bacteroidetes (3). bHLH35 proteins are transcription factors induced by effector-triggered immunity (ETI), and also involved in tolerance to abiotic stresses[51]. The SNP *si1:32157654* located in the generic region of WAK2 (wall-associated receptor kinase 2) was associated with 30 common OTUs, including Acidobacteria (21), Bacteroidetes (4), Proteobacteria (4) and Actinobacteria (1). The WAK2 protein bound to pectin, is required for cell expansion and is induced by a variety of environmental stimuli, including pathogens and wounding[52]. Similarly, the 50 common OTUs were found to be associated with FLS2 (*si7:2994337*, Supplementary Data 9), a flagellin sensor that perceives conserved microbial-associated molecular patterns (MAMPs) in the extracellular environment[53]. *Clostridia* OTU_19207 and *Nocardioides* OTU_26357 associated *si8:20598566* located within the gene of NPF1.2 (Fig. 5 and Supplementary Data 9), which is involved in ABA importing and nitrate utilization, regulates plant development and influences the root microbiota[14,35,54]. An NPF1.2 homologue in loci *si1:20064466* was significantly associated with Bacillaceae OTU_28839. Collectively, host genes related to plant immunity (RPM1, RGA2, HSL1, CRKs, LRR-RLKs), metabolites (Flavonoids, Diterpenes, amyA, alpha-N-arabinofuranosidase, beta-glucuronosyltransferase), nutrient uptake (Acid phosphatase, $Mg^{2+}$ transporter, $H^+$-transporting ATPase), plant hormone signal transduction (BRI1, DELLA protein, EFR3, PI-PLC, SDR, ARR1) and others (E3 ubiquitin protein ligase) are perhaps common host genetic factors with function to modulate root microbial composition assembly (Fig. 5 and Supplementary Data 9).

Plants primarily influence their microbiomes through targeted interactions with key taxonomic groups or diffuse interactions with entire communities[55]. To further investigate the mode of host-microbe interactions, the hub microbial taxa and non-hub microbial taxa and their associated genes were identified. Firstly, we defined hub taxa as OTU with high values of degree (>400) and closeness centrality (>0.5) in the network as described in a previous study[56], resulting in 102 hub OTUs. We identified that 90 hub OTUs and 748 non-hub OTUs had significant associations with the host genetic SNP loci (Supplementary Fig. 10A, Supplementary Data 9), indicating host plant might interact with these hub microbes and diffusely interact with these non-hub microbes. We aggregated these SNP-associated hub OTUs (90 hub OTUs) and non-hub OTUs (748 OTUs) into 12 and 36 microbial orders, respectively. Comparative analysis showed that one order GP7 was only composed of SNP-associated hub OTUs, and 25 orders such as

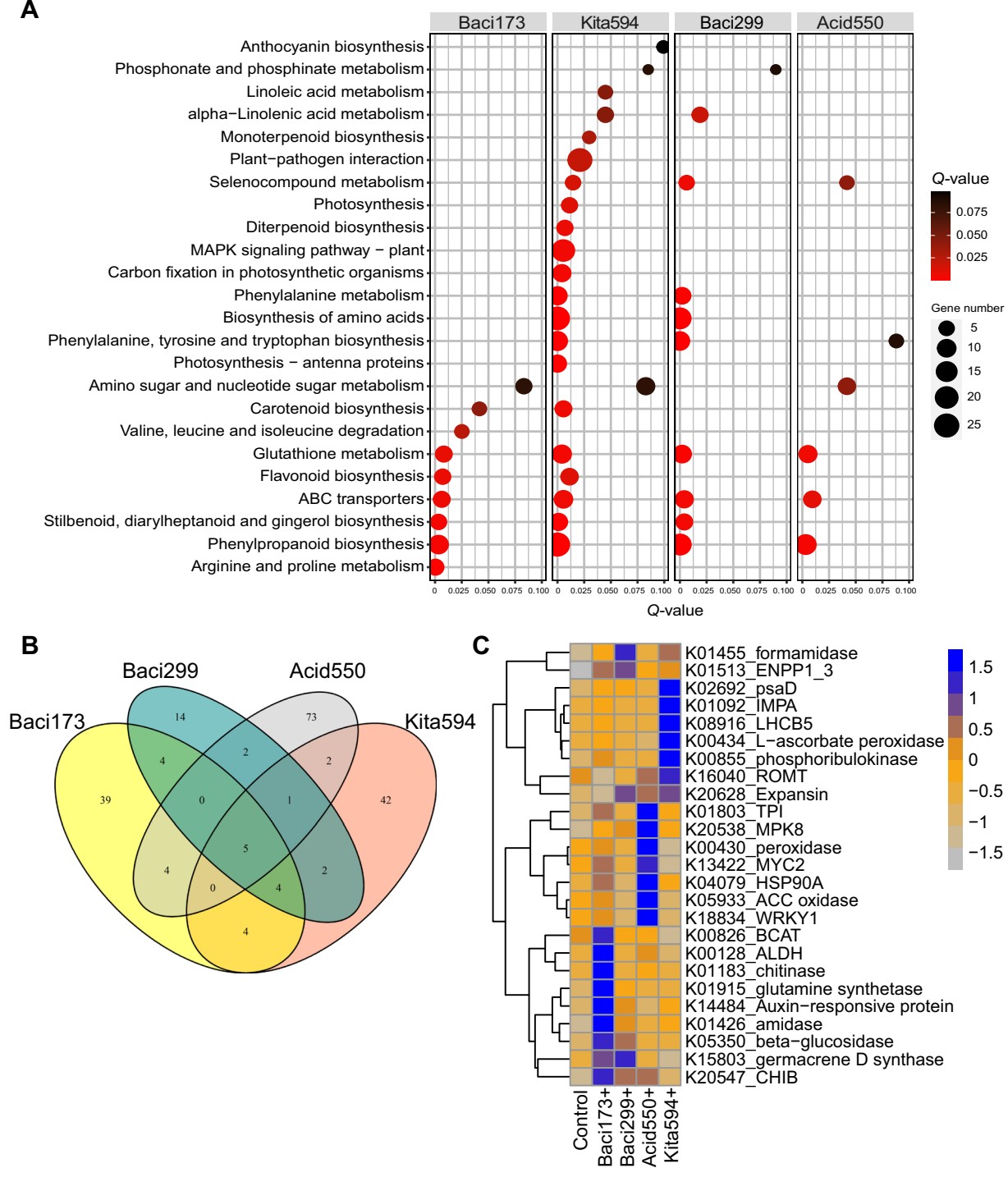

**Fig. 4 | Marker strains induce distinct genes in foxtail millet. A** KEGG enrichment analysis of differentially expressed genes in Baci173-, Baci299-, Acid550-, Kita594- inoculated seedlings. The differentially expressed genes represent the genes that were significantly upregulated or downregulated in seedlings inoculated with marker strain compared with control. Circle size represents the number of genes within the pathway and color represents the significance of the pathway. **B** Venn diagram showing the overlap of the significantly upregulated genes under different inoculations. **C** Transcript abundance of genes that were induced only in Baci173-, Baci299-, Acid550- and Kita594-inoculated seedlings, respectively.

Sphingobacteriales, Bacillales, Ohtaekwangia, Sphingomonadales and Acidimicrobiales were only composed of SNP-associated non-hub OTUs, and 11 orders were composed of both SNP-associated hub and non-hub OTUs (Supplementary Fig. 10B). These data indicated that the foxtail millet employed two modes to structure the rhizoplane microbiota: targeted interaction with several hub microbes and diffused interaction with most of the microbes. To decipher the potential mechanism of the interaction between plant and microbe, the candidate host genes around the SNP loci associated with the hub and non-hub OTUs were extracted separately. The networks showed that the host immune genes FLS2 and transcription factor bHLH35 are widely associated with the hub and non-hub taxa (Supplementary Fig. 11A, B). However, the host plant still employed different genes to interact with different taxa (Supplementary Fig. 11C), suggesting a taxa-dependent regulation model.

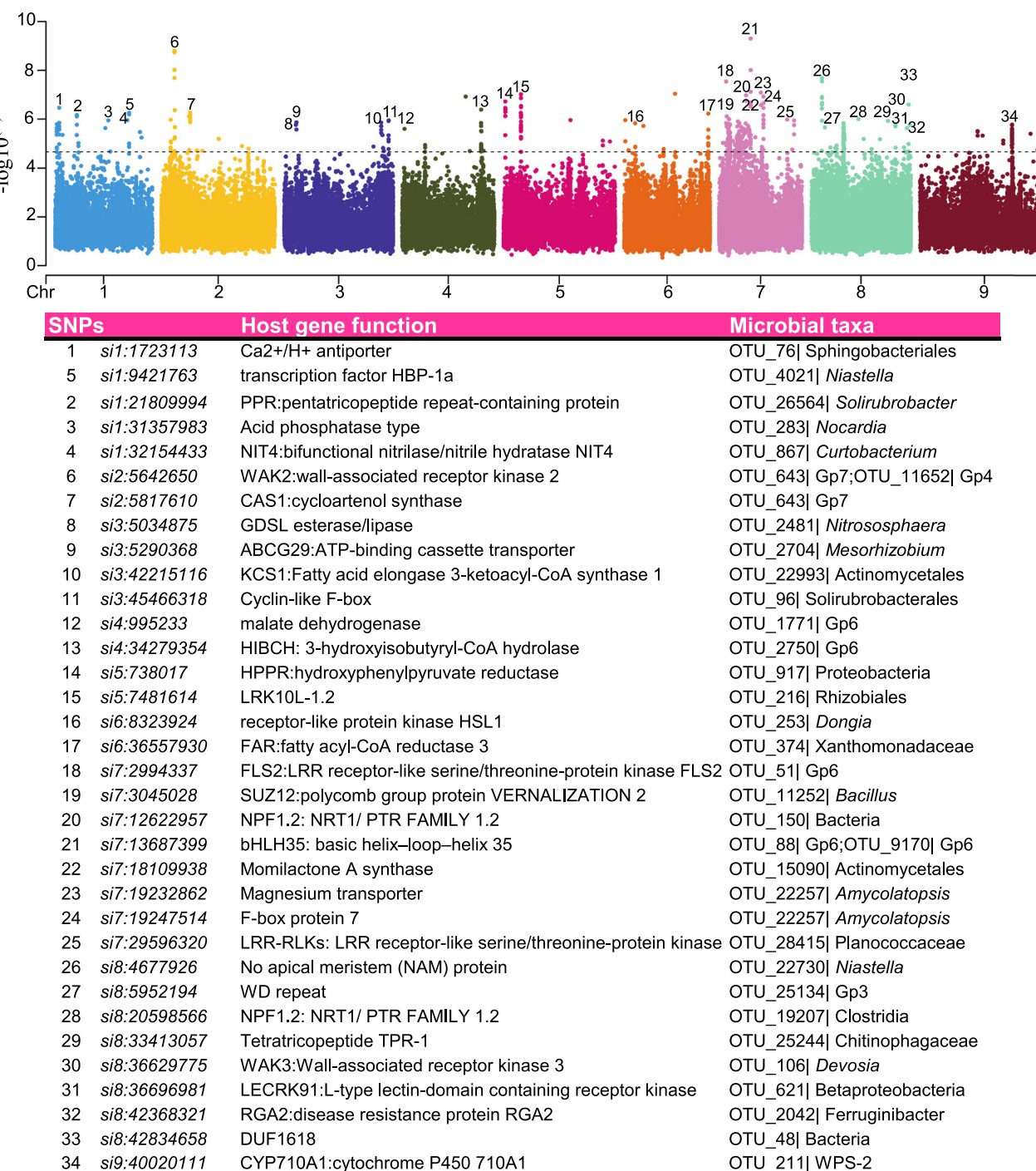

**Fig. 5 | Host genetic variation correlated with common bacterial taxa.** Manhattan plots show the significant SNPs for microbial abundance. SNPs located in gene coding regions are labeled with numbers. Details of the associations between the host genes and microbial species are given in the table below. All of these associations of SNP loci and microbial OTUs were significantly lesser than 2.01e−5.

## Microbe-mediated growth depends on plant genotype

To determine if the genotype-dependent rhizoplane microbiota influence agronomic traits in foxtail millet, we compared the 838 SNP-associated OTUs (mGWAS identified) with the 257 marker OTUs (MWAS identified). We discovered that 219 of the SNP-associated OTUs overlapped with the marker OTUs in our data sets, covering 85.2% of 257 marker OTUs. (Supplementary Fig. 12A, 219 out of 257 = 85.2%). 682 SNP loci were significantly associated with 219 marker OTUs (here called marker OTU-associated SNPs). However, for the 682 marker OTU-associated SNPs, only 4 overlapped with the 45 non-redundant marker SNPs (GWAS identified) that were associated with the aforementioned agronomic traits of foxtail millet (Supplementary Fig. 12B). Most of the genetic variations that were associated with marker OTUs were not directly associated with agronomic phenotypes. These genetic variations might affect agronomic phenotypes indirectly, only in the presence of environmental factors such as marker microbes. Moreover, the Mantel test also showed that SNP-associated marker OTUs had higher correlations with the growth trait (MSPD and MSW) than non SNP-associated marker OTUs, while having no difference in correlations with trait TSLW, MSPW, PGW and MSPL (Supplementary

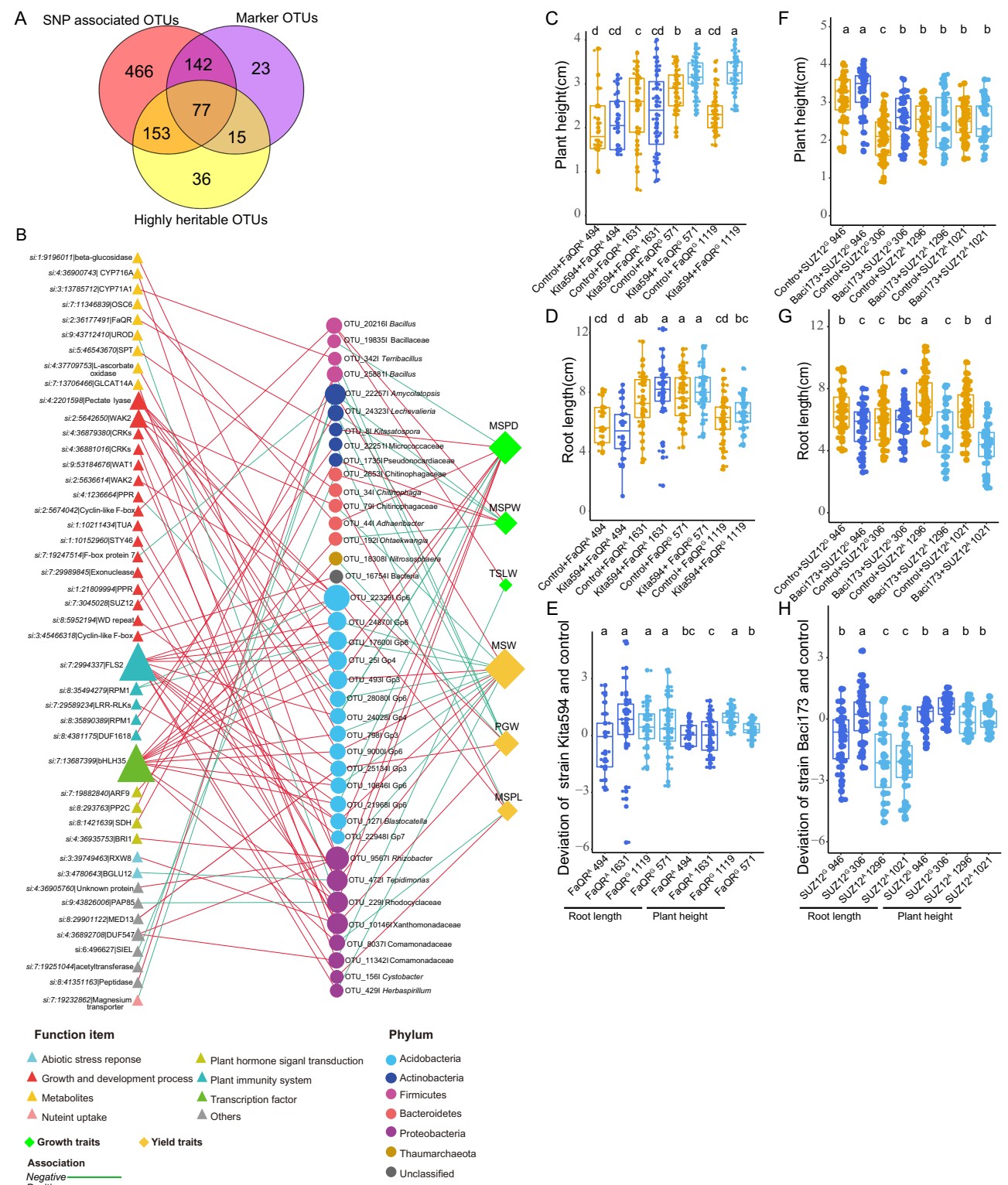

Table 2). It means that the genotype-dependent marker OTUs might explain more variances in plant growth traits.

To decipher host plant genetic mechanisms for marker microbe selection, KEGG pathway enrichment analysis revealed that the genes within or nearby the significant SNP loci were enriched in pathways related to plant-pathogen interaction (ko04626), MAPK signaling (ko04016), Steroid biosynthesis (ko00100) and so on (Supplementary Data 10). Specifically, the genes enriched in plant-pathogen interactions included microbial pattern-recognition receptors (PRRs),

disease-resistant genes RPM1 and RPS2, an activator of pathogenesis-related genes PTI1 and PTI6, key regulators of plant immune responses CALM and transcription factor WRKY25. These results suggest that the plant defense genes may also underpin the microbial ecology in the root habitat in addition to protecting from pathogens.

Among the 219 SNP-associated marker OTUs, 77 were highly heritable (Fig. 6A). The association between host genetic variation, the abundance of specific marker microbes and phenotypes, especially for 77 genomic heritable marker OTUs were closely examined (Fig. 6A,

**Fig. 6 | Marker strains influence foxtail millet phenotypes in a genotype-dependent manner. A** Venn diagram displaying the overlaps among 838 SNP associated OTUs, 257 marker OTUs and 281 highly heritable OTUs. **B** A network of associations between the candidate genes and marker microbial OTUs. Edges between the marker OTUs and host genes were colored according to the correlation coefficients. The pink color represents the positive correlations while the green color represents the negative correlations. The circle represents the OTUs colored according to the phylum taxonomy information, the triangle represents the genes colored according to the function module information, the square represents the growth traits colored in green and the yield traits are shown in yellow. Plant height (**C**) and root length (**D**) of seedlings of FaQR reference cultivars (C494 and C1631) and allele cultivars (C571 and C1119) grown axenically (no bacteria, control) or with growth-promoting Kita594. $n = 22, 26, 36, 37, 50, 46, 45$ and 41(from left to right) biological replicates. Plant height (**F**) and root length (**G**) of seedlings of SUZ12 reference cultivar (C946 and C306) and allele cultivar (C1296

and C1021) grown axenically (no bacteria, control) or with growth-suppressing Baci173. $n = 39, 32, 46, 40, 50, 24, 45$ and 31 (from left to right) biological replicates. The deviation of promoting and suppressing effect of marker strain Kita594 (**E**) and Bci173 (**H**) were calculated separately. $n = 26, 37, 46, 41, 26, 37, 46$ and 41 (from left to right) biological replicates for the treatment with marker strain Kita594 (**E**). $n = 32, 40, 24, 31, 32, 40, 24$ and 31 (from left to right) biological replicates for the treatment with marker strain Bci173 (**H**). Different letters in **C** to **H** indicate statistical significance (adjusted $P < 0.05$) among the treatments according to one-way ANOVA and LSD test at the 5% level. In **C**, $df = 7$, $F = 19.73$, adjusted $P < 2.0e{-}16$; in **D**, $df = 7$, $F = 13.76$, adjusted $P = 2.09e{-}15$; in **E**, $df = 3$, $F = 2.10$, adjusted $P = 0.102$; $df = 3$, $F = 18.29$, adjusted $P = 4.04e{-}10$; In **F**, $df = 7$, $F = 18.11$, adjusted $P < 2.0e{-}16$; in **G** $df = 7$, $F = 14.83$, adjusted $P < 2.0e{-}16$; in **H**, $df = 3$, $F = 19.57$, adjusted $P = 2.0e{-}10$; $df = 3$, $F = 6.751$, adjusted $P = 2.90e{-}4$. The box edges depict the 75th and 25th percentiles, respectively and the line within the box represents the median. The whiskers extend 1.5 times the IQR from the top and bottom of the box, respectively.

Supplementary Data 8). Remarkably, plant defense-related genes and transcription factors, such as the plant immune receptor FLS2 (*si7:2994337*), transcription factor bHLH35 (*si7:13687399*) and WAK2 (*si:2:5642650*) had a dominant impact on the marker OTUs from the phylum of Acidobacteria (Fig. 6A). In contrast, genes involved in nutrient uptake, metabolites and abiotic stress response, such as magnesium transporter (*si7:19232862*), triterpene synthase (*si:7:11346839*, achilleol B synthase), BGLU12 (*si:3:4780643*, Beta-glucosidase 12) and RXW8 (*si:3:39749463*, CSC1-like protein RXW8), mainly associated with marker OTUs from Actinobacteria, Bacteroidetes and Proteobacteria, which mostly contribute positively to the growth and yield traits of foxtail millet (Fig. 6A). Other genes involved in plant growth and development processes, such as SUZ12 (Polycomb protein SUZ12) and WAT1 (WAT1-related protein), impacted the marker OTUs from Firmicutes (Bacillaceae OTU_19835) and Proteobacteria (Xanthomonadaceae OTU_10146) respectively, but these marker OTUs have opposite effects on the growth of foxtail millet (Fig. 6A). Additionally, we observed strong associations between the positive marker *Acidovorax* OTU_46 and EREBP-like factor (*si7:27291504*, dehydration-responsive element-binding protein 1B-like), and between positive marker *Kitasatospora* OTU_8 and FaQR (*si2:36177507*, 2-methylene-furan-3-one reductase) (Fig. 6A). To explore the host genetic mechanisms that might drive the associations of the plant host gene and rhizoplane microbiota, we examined the specific expression pattern of candidate genes from the RNA-seq datasets obtained from the sterilized soil experiments. Obviously, the genes FaQR, vWA (von Willebrand factor, type A), SUZ12 and EREBP-like factor (ethylene response element binding protein) exhibited significant variation after being inoculated with strain Kita594 (*Kitasatospora* OTU_8), Baci299 (Bacillaceae OTU_22228), Baci173 (Bacillaceae OTU_19835) and Acid550 (*Acidovorax* OTU_46) compare to control, respectively (Supplementary Fig. 13), implying that the candidate host genes likely interacted with specific bacterial strains.

Finally, based on cultivars with different genotypes, the influence of functional SNPs on marker OTU abundance was thoroughly examined. The abundance of marker OTUs shifted among the different genotypes at the most strongly associated SNPs (Supplementary Fig. 14). We hypothesize that host gene-regulated promotion/suppression microbes could establish genotype-dependent microbe-mediated growth phenotypes. To test this hypothesis, we germinated the FaQR and SUZ12 reference and allele foxtail millet cultivars on sterile plates inoculated with a growth-promoting or suppressing strain that corresponds to each cultivar: the growth-promoting strain Kita594 to FaQR reference (C494 and C1631) and allele (C1119 and C571) genotype cultivars, the growth-suppressing strain Baci173 to SUZ12 reference (C946 and C306) and allele (C1021 and C1296) genotype cultivars. Intriguingly, we found that strain Kita594 had a statistically significantly shoot-promoting effect only on the allele cultivars, but not on reference cultivars (Fig. 6C–E, adjusted $P < 0.05$ by

ANOVA-LSD), supporting that plant-growth promoting rhizobacteria support genotype-dependent cooperation with the plant. We observed strong root growth inhibition in seedlings inoculated with the growth-suppressing strain Baci173 (Fig. 6F–H, adjusted $P < 0.05$ by ANOVA-LSD), and a more significant suppressing effect on root length was observed in the allele cultivars (C1296 and C1021) compared to the reference cultivars (C946 and C306). Significant effects of the interaction between the genotype and strain Kita594 and strain Baci173 on the shoot and root length were also detected by PERMANOVA, respectively (genotypes*Kita594: $R^2 = 13.048$, $P < 0.001$; genotypes*Baci173: $R^2 = 0.07$, $P < 0.001$, Supplementary Table 3). Together, these results suggest that host genetic variation might impact the interactions between marker strains and host plants, finally affecting the plant phenotypes.

## Discussion

Although GWAS is the most popular approach to date to explain the underlying mechanism of plant phenotypes, it remains limited to capturing complex agronomic traits. MWAS has emerged as a new approach to address this challenge. MWAS on human cohorts has enabled the mining of gut microbial markers for complex traits, including obesity and type 2 diabetes (T2D)[57]. However, to date, very few MWAS have been conducted in plants[26,58]. By leveraging the information obtained from the rhizoplane microbiota of the 827 foxtail millet cultivars, coupled with the genome-wide-association summary statistics for millet growth and yield traits, most of the variations in TSLW, MSPD, and MSPW traits were predicted by rhizoplane microbiota alone, suggesting the important effects of rhizoplane microbiota on host phenotypes. Thus, it will be important to consider MWAS in studies aiming to improve crop agronomic desirable traits.

We further validated the growth-promotion and -suppression effects of microbial markers predicted by MWAS, supporting that MWAS can be used to discover rhizobacteria that modulate plant growth. The pathways induced by all marker strains were mainly related to metabolites, hormone biosynthesis (e.g., ABA), immune responses and nutrient uptake, suggesting these functions generally regulate plant-microbe interactions. Auxin and auxin-responsive genes that control plant root and shoot growth were induced by suppressing strains, consistent with recent observations[59]. The suppressing strain also induced branched-amino acid and L-glutamine, which inhibit root and seedling elongation[43,44]. In addition, the genes induced by growth-promoting bacteria varied among marker strains, indicating that plants have customized responses to different bacteria.

The heritable taxa from foxtail millet strongly overlapped with sorghum and maize, implying a similar co-evolution pattern of plant and their microbiota across hosts. mGWAS was applied to reveal the genetic loci correlated with rhizoplane microbial abundance, which has been only conducted in *Arabidopsis thaliana*[18] and *Sorghum bicolor*[45]. Numerous potential genes governing the microbial effect in

this study were at first excavated. Notably, among the 257 marker microorganisms, 219 were associated with the host genetic variations. The direct GWAS analysis of the relationship between the corresponding SNPs and millet traits were mostly not detected at the suggestive genome-wide significant cutoff. The marker OTU-associated SNPs did not overlap with marker SNPs, suggesting that host genetic variations and the microbiota affect the plant phenotypes independently. Moreover, the SNP-associated marker OTUs had higher correlations with plant growth traits than non SNP-associated marker OTUs. These results indicate that host genetics might shape the composition of the root microbiota which in turn shapes the agronomic traits of foxtail millet, similar to the Mendelian randomization relationships among the gut microbiome, short-chain fatty acids and metabolic diseases[60].

We further experimentally validated the genotype-dependent microbe-mediated growth promotion of foxtail millet in cultivars, SUZ12 and FaQR. We found the promoting or suppressing functions of the target strains in foxtail subspecies were associated with the cultivar genotypes. The differences in the field environments may cause distinct loci or host processes to shape the root microbiota. Independent GWAS in multiple environments or single environment could have both the same and different association features, but with different emphasis. In this study, numerous potential genes associated with the root microbiota were identified by mGWAS based on 827 foxtail millet cultivars in a single environment, and changes in effects (genotype-by-environment interactions) are consistent across the environments. Among these, the genes such as FLS2, WAK2 and GDSL-type lipase have been reported to function in structuring the plant microbiota assembly or mediate pathogen and stress responses[11,52,61]. Of course, numerous genes governing the microbial effect in this study were excavated for the first time and need yet to be proved by further experiments. In the future, independent GWAS performed in multiple environments will dissect shared genetic loci which affect root microbial communities. Our study outlines a reciprocal interplay among host genetic variations, root microbiota and crop agronomic traits. We propose that the exclusive coupling of plant-bacteria interactions in agricultural practice, or precision microbiome engineering, will be crucial for efficient and sustainable agriculture.

## Methods

### Study samples
Thousands of foxtail millet cultivars were planted in the natural fields at Yangling Agricultural Hi-tech Industrial Demonstration Zone (34°16′18″ N / 108°4′59″ E, Shanxi, China) in September 2013, with 20 repeats for each cultivar. For each cultivar, an extensive dataset of root associated microbiota and agronomic traits information was collected and has been described in detail previously[26]. Genotyping data of 1127 cultivars were generated by restriction site associated DNA sequencing (RAD sequencing) as described in the previous study[62]. Briefly, genomic DNA was extracted from young leaves using a standard cetyltrimethylammonium bromide (CTAB) method. The genomic DNA of each sample was digested with TaqI which recognizes the sequence T^CGA. The unique index was then added to each sample and 400–600 bp DNA fragments were selected and amplified. Samples were sequenced on Illumina HiSeq2000 platform, generating 90-bp paired-end reads.

### Variant calling, evaluation and annotation
The raw reads were filtered using SOAPnuke (v.1.5.6) based on the percentage of low-quality bases and ambiguous bases (Ns) or adapter sequences with parameter (-l 5 -q 0.5 -n 0.03-f GATCGGAAGAGCAC ACGTCTGAACTCCAGTCAC-r-AGATCGGAAGAGCGTCGTGTAGGG AAAGAGTGTA). Clean reads were mapped to the *Setaria italica* cv. Zhanggu reference genome (v.2.3) (https://ftp.cngb.org/pub/CNSA/ data2/CNPhis0000549/Foxtail_millet/)[27] using BWA-MEM (v.0.7.12-r1039)[63] with the parameter (-M -R). The generated SAM files were converted into BAM files using SAMtools (v.0.1.19–44428 cd)[64], and then sorted by reference position using SortSam.jar in the Picard package (v.1.54, http://broadinstitute.github.io/picard/). To improve the accuracy of alignment around InDels, realignment was conducted using RealignerTargetCreator and IndelRealigner in GATK (v.3.6)[65] with default parameters. SNPs were then called using HaplotypeCaller and filtered using VariantFiltration with parameter (-genotypeFilter-Expression DP < 3.0 -genotypeFilterName lt_3 -setFilteredGtToNocall -filterExpression QD < 2.0 || FS > 60.0 || MQ < 40.0 || MQRankSum < −12.5 || ReadPosRankSum < −8.0). Only bi-allelic SNPs were kept using bcftools (v.1.2, http://github.com/samtools/bcftools). The imputation step was performed using BEAGLE (v.4.1) with default parameters[66]. The SNP markers with an estimated allelic R-squared value ($AR^2$) of more than 0.3 were retained for subsequent analysis.

All pairwise identity-by-state (IBS) distances between individuals were calculated for the filtered SNP data using PLINK (v.1.90) with the command (plink -genome)[67]. To remove duplicated or related individuals, we kept only one with a high calling rate between two individuals with IBD of more than 0.185. To further control population stratification, we performed multidimensional scaling (MDS) analysis on the genome-wide IBS pairwise distances and removed outlier individuals, and finally, 827 samples remained. The SNPs were further kept with minor allele frequency (MAF) ≥ 0.01, genotype calling rate >0.7 and individual-level genotype calling rate >0.7 with the command (plink -maf 0.01 -geno 0.3 -mind 0.3). The genomic location and putative function of SNPs were predicted according to the gene model on the *S. italica* reference genome using snpEff (v.4.3t)[68].

### GWAS on the phenotype of growth and crop yield
Twelve agronomic traits namely top second leaf length (TSLL), top second leaf width (TSLW), main stem height (MSH), main stem width (MSW), panicle diameter of the main stem (MSPD), fringe neck length (FNL), panicle length of the main stem (MSPL), per plant grain weight (PGW), main stem panicle weight (MSPW), hundred kernel weight (HKW), spikelet number of the main stem(MSSN) and grain number per spike(SGN) were recorded for 827 foxtail millets (Supplementary Data 1). To improve the power of GWAS and avoid confounding factors, the GWAS was conducted based on a linear mixed model with correction of the first ten principal components and kinship using GEMMA (v.0.98)[69]. To determine the genome-wide significant cutoff for GWAS results, we estimated the number of genome-wide effective SNPs by pruning SNPs within 500 bp and with an $R^2 ≥ 0.2$ by a slide window approach with a window size of 500 bp and step of 100 bp using PLINK, and the number of effective SNPs were determined to be 49,512. We then selected 2.01e−5 (1/effective SNP number) as the suggestive genome-wide significant cutoff, resulting in 13, 18, 8, 20, 5, 3, 15, 3,1 and 5 significant loci for MSPD, MSW, TSLL, FNL, MSH, PGW, MSPL,HKW, MSSN, SGN, respectively (Supplementary Data 2). No significant locus was observed for TSLW and MSPW. Due to the lower density of SNPs in this study, genes located within 20 kb or high linkage disequilibrium regions ($R^2 > 0.4$) around the significantly associated SNPs were identified as the candidate genes. Enrichment of Gene Ontology (GO) term and KEGG pathway were performed for candidate genes according to EnrichmentPipeline (v.1.01).

### Modeling the effects of host genotype and root microbiota on phenotypes by MWAS
To evaluate the impact of host genetic SNPs on plant growth and yield traits, host genetic SNPs were used as input data of the linear regression model and 10 genetic PC were used as covariables. The linear regression model (Eq. 1) was used to assess host genetic SNPs against the trait TSLW, MSW, MSPD, MSPW, PGW and MSPL, separately (R

package v.4.0.2):

$$Z \sim AX + E \quad (Z \text{ is the value of the phenotypic trait}, X \text{ is the genotypes of effective SNPs})$$
(1)

For each trait, the effective SNPs ($P < 1.0e{-}4$) served as input data. Five-fold cross-validation was performed to estimate the ability of the model to predict new data. The original 827 samples were randomly partitioned into five equal-sized groups, of which four groups were randomly selected to train the prediction model and the remaining one group was used to test the model fitness. The cross-validation process is then repeated five times, with each of the five sub-group samples used exactly once as the validation data. Specifically, the prediction model selection for each trait was performed according to the Akaike's Information Criterion (AIC) value based on the training data set. The fitness of the selected model was then estimated according to correlation efficiency ($R^2$) of the observation and prediction value of the trait in the testing data set. The predictive models generated from different sample groups show different performances in the prediction of the testing data. To reduce the noise in the estimated model performance, we repeat the five-fold cross-validation process 30 times and report the mean performance across all folds and all repeats.

To assess the influence of root microbiota on plant growth and yield traits, rhizoplane microbiota was used as input data of the linear regression model and 10 genetic PC were used as covariables (Eq.2) (R package version 4.0.2):

$$Z \sim BY + E \quad (Z \text{ is the value of the phenotypic trait, Y is the count of rhizoplane OTUs})$$
(2)

The OTU table of 827 rhizoplane samples was extracted from the 16 S variable region V4-V5 dataset of root microbiota of foxtail millet[26]. Then 1004 common OTUs having an occurrence frequency higher than 70% were extracted. Among these, the top 200 OTUs along with significant positive or negative correlation with the trait of foxtail millet (adjusted $P < 0.05$) were selected as input of the linear regression model. Then five-fold cross-validation was performed and 827 samples were randomly divided into five equal-sized groups, of which four groups were randomly selected to train the prediction model according to the Akaike's Information Criterion (AIC) value and the other samples were used to test the fitness. To reduce the noise in the estimated model performance, we repeat the five-fold cross-validation process 30 times and report the mean performance across all folds and all repeats.

Similarly, the combined effects of host genetic SNPs and root microbiota on plant traits were calculated by the mixed linear model, and 10 genetic PC were used as covariables (Eq. 3):

$$Z \sim AX + BY + CXY + E \quad (Z \text{ is the value of phenotypic trait};$$
$$X \text{ is the genotypes of effective SNPs}; Y$$
$$\text{is the counts of rhizoplane OTUs})$$
(3)

For each trait, the effective SNPs served as input data (adjusted $P < 1.0e{-}4$) and microbial OTUs from the model mentioned above were screened as the input data of the mixed linear model. We firstly added the effective SNPs in the model to stepwise regress against the phenotype, then the microbial OTUs were added, followed by interactions of SNPs and OTUs. Similarly, five-fold cross-validation was performed using the same sample groups as above. To reduce the noise in the estimated model performance, the five-fold cross-validation process was repeated 30 times and reported the mean performance across all folds and all repeats. The predictive models with the best prediction accuracy for the phenotypes using the SNP and OTU variables were

selected, resulting in the identification of marker OTUs (Supplementary Data 4 and 5).

## Definition of abundant and rare microbial OTUs
We defined abundant or rare OTUs according to the previously suggested rules[70,71]. Briefly, the OTUs with relative abundances ≥0.1% across more than 50% of samples were defined as abundant OTUs (AT), whereas the OTUs with relative abundances <0.01% across more than 30% of samples but never abundant (≥0.1%) more than 30% samples were defined as rare OTUs (RT). Those OTUs neither belonging to abundant taxa nor rare taxa were defined as moderate OTUs (MT). The distributions of AT, MT and RT in whole and common-sub communities were analyzed. The rare abundant OTUs (RT) have most of the OTU numbers (81.20%) in the whole community, but covered an average 39.26% sequence abundance, similar to the previous study[70]. The moderate OTUs have the highest numbers (83.67%) in the common-sub community, followed by abundant OTUs (12.85%) and rare OTUs (3.48%) (Supplementary Table 1).

## The AVD value calculation
The community stability was assessed using the average variation degree (AVD) index, which is used as an indicator of soil microbiota stability[72]. AVD is calculated using the deviation degree from the mean of the normally distributed OTU relative abundance using the following equation (Eq. 4).

$$\text{AVD} = \frac{\sum_{i=1}^{n} \frac{|x_i - \bar{x}_i|}{\delta_i}}{k \times n}$$
(4)

($x_i$ is the rarefied abundance of the OTU in one sample, $\bar{x}_i$ is the average rarefied abundance of the OTU in one sample group, and $\delta_i$ is the standard. $k$ is the number of samples in one sample group, $n$ is the number of OTUs in each sample group). A lower AVD value indicates higher microbiota stability. To compare the structure stability with the common OTU community, the OTU sub-tables with 30 and 50% occurrence frequency were extracted from the whole communities, and the AVD values were calculated according to the above equation, respectively.

## The microbial co-occurrence network analysis
The network based on Spearman correlation scores was calculated using 'corr.test' function in R package lsr 0.5.2 and only robust (Spearman's $r > 0.4$ or $r < -0.4$) and statistically significant (adjusted $P < 0.05$) correlations were kept. The topology characteristics (e.g., degree, betweenness centrality, closeness centrality and hub score) of the network were calculated using R package igraph 1.2.11 to quantitatively describe the species' role in communities. The significance of the differences between the groups was detected using Kruskal-Wallis rank sum test (one-way) with $P < 0.05$. The "hub species" was defined as OTU with high values of degree (>400) and closeness centrality (>0.5) in the network[56]. Thus 102 hub OTUs were identified in 1004 common OTUs.

## Heritability calculations
To estimate OTU heritability, their counts of the 1004 common microbes in each sample were normalized by the CSS (cumulative sum scaling) method. Relative abundance was log-transformed to fit a normal distribution and genetics complex trait analysis (GCTA) (v.1.92.2) was used to estimate the phenotypic variance explained by all SNPs with a genome-based restricted maximum likelihood (GREML) method[73,74]. Adjustment of the $P$-value for all 1004 OTUs was performed using the Benjamini-Hochberg algorithm (BH). A total of 281 highly heritable microbes were observed, giving $H^2$ values in the range of 0.15 to 0.32.

## mGWAS on the root associated microbiota

To evaluate the impact of host genetic variations on root microbiota abundance, we ran mGWAS on 1004 common microbes of foxtail millet. To make the CSS value to be normally distributed, the rank-based inverse normal transformations (Rank-based INT) were applied to transform the distribution of each OTU to make it appear more normally distributed[75]. Rank-based INT entails creating a modified rank variable and then computing a new transformed value of the phenotype for the $i$th subject. A detailed description of the calculating formula is given below (Eq. 5):

$$y_i^t = \Phi^{-1}\left(\frac{r^i - c}{N - 2c + 1}\right) \tag{5}$$

where $r^i$ is the ordinary rank of the $i$th case among the N observations and $\Phi^{-1}$ denotes the standard normal quantile (or probit) function. The value of $c = 3/8$ is recommended by Blom[76]. The mGWAS of normalized OTU abundance was performed based on a linear mixed model with correction of the first ten PCs and kinship using GEMMA (v.0.98). The Q-Q plot of the mGWAS results was generated to detect the existence of the systematic bias. SNPs with a suggestive $p$-value $<2.01e-5$ were considered to have putative associations with microbes.

2108 significant SNP loci associated with 838 OTUs were identified (Supplementary Data 9). The genes located within 20 kb or high linkage disequilibrium regions ($R^2 > 0.4$) around the significantly associated SNPs were identified as the candidate genes, and the GO term and KEGG pathway enrichment for these genes were also performed.

## Isolation of root-derived bacteria

The 2000 cultivars of foxtail millet were re-planted in Yangling, China in 2018. For each cultivar, roots were collected from three plants and pooled. All of the samples were stored at −20 °C and immediately taken back to the laboratory for further preparation. The root samples were cut into small sections of 2–3 cm, put into a 2-ml Eppendorf tube containing 1.5 ml sterile PBS-S buffer and washed on a shaking platform for 20 min at 25 r/s. The washing buffer was then subjected to centrifugation for 5 min at $137 \times g$. The supernatants were diluted from $10^{-1}$ to $10^{-7}$ and then $10^{-4}$ and $10^{-6}$ dilutions were distributed and cultivated in 96-well microtiter plates in Nutrient Agar (NA) media (BD Company) for 48–72 h at 28 °C. After purification by three consecutive platings on solidified media, 644 bacterial clones were obtained. The full length of 16 S $rRNA$ gene for each strain was amplified with bi-barcoded universe primers (27 F 5′-GAGTTTGATCCTGGCTCAG-3′; 1492 R 5′-TACCTTGTTACGACTT-3′) (Supplementary Table 4). The PCR products of 322 strains were pooled together in equimolar ratios and sequenced on a nanopore platform. The sequencing reads were base-called using guppy. After trimming the low-quality reads, 257 bacterial strains with distinct 16 S rRNA gene sequences were retained. The sequences of 257 marker OTUs were aligned with 16 S rRNA gene sequences of these strains by Usearch (v.10.0.240). The representative stain of each OTU was identified with the best hit of 16 S rRNA gene with gene similarity of more than 97%, resulting in 24 marker OTUs with representative strain.

## Plant growth promotion assay

To validate the promoting or suppressing effect of marker microbes on foxtail growth, 10 marker OTUs with representative strain and with top beta estimation in the regression model were selected for the validation experiment, including positive marker OTUs: *Acidovorax* OTU_46 to the strain Acid550, Bacillaceae OTU_22228 to strain Baci299, *Kitasatospora* OTU_8 to strain Kita594, *Bacillus* OTU_19414 to strain Baci154, *Bacillus* OTU_25704 to strain Baci312, Bacillales OTU_381 to strain Baci429, and negative marker OTUs: *Shinella* OTU_37 to strain Shin228, *Bacillus* OTU_54 to strain Baci81, Bacillaceae OTU_19835 to strain Baci173 and Bacillaceae OTU_28133 to strain Baci554. First, the

representative strain for each marker OTU was purified by three consecutive platings on the respective solidified media and an individual colony was cultured in 50 ml tubes with NA liquid medium at 28 °C for 5 d. The optical density $OD_{600}$ of strains was adjusted to 0.5 and cells were collected by centrifugation under $2653 \times g$ for 10 min. The pellet was dissolved with an equal amount of sterilized water. Huagu12 is a foxtail millet (*Setaria italica*) cultivar bred by BGI Institute of Applied Agriculture (Shenzhen, China). The seeds of foxtail millet Huagu12 were sterilized with 0.1% NaClO for 5 min, washed with sterilized water 5 times, and transferred into plates with sterilized filter paper. For each treatment, 2 ml bacterial suspension was added to the plate with sterilized seeds, and three repeats were included. The control was added with 2 ml of sterilized water. Seeds were grown at 22 °C, 12-h light/12-h dark and 21% humidity. After a 7-day co-culture, plant height and root length were measured to evaluate plant growth conditions.

To further validate the promoting or suppressing effect of marker microbes on foxtail millet growth conditions, we sowed the germfree Huagu12 seeds into the small basin with the sterilized soil collected from the planting field of foxtail millet in the greenhouse of CNGB, Shenzhen. Then the suspension of marker strain was used to water the 5-day old seedlings in the basin, with three repeats for each strain. The control was watered with sterilized water. After 14 days of inoculation, the plant height and root length were measured and assessed by t-test with $P$-value adjusted by the BH method.

The foxtail millet seeds with reference genotype and allele genotype were sterilized and put on the filter paper in the germ-free plate, respectively. Then the 2 ml suspension of marker strain was inoculated into the plate. The control was added with 2 ml of sterilized water. After 7-days of co-culture, the difference in root length and plant height compared with control were determined by ANOVA with LSD test. The promoting effect and suppressing effect on the root and plant height were evaluated by subtracting the average of the control. The difference between the effects of the strain Baci173 and Kita594 on reference genotype cultivars and mutant genotype cultivars were detected by Wilcoxon rank test, respectively. The impacts of genotype and strain on seedling's height and root length were calculated with adonis function in the R package (v.4.0.3).

## RNA extraction and sequencing

As strains Kita594, Baci299, Acid550 and Baci173 exhibited significant effects on foxtail millet phenotypes, their inoculated seedlings and control seedlings were selected to perform transcriptomic sequencing, separately. Five seedlings of foxtail millet from each treatment of the sterilized soil experiment were pooled and flash frozen and stored at −80 C until processing. RNA from the foxtail millet was extracted using Column Plant RNAout Kit according to the instructions (TIANDZ, China). The RNA extraction was quantified and assessed for integrity using the NanoDrop (Thermo, USA) and 2100 Agilent Bioanalyzer (Agilent, USA) before subsequent experiments. The 1ug qualified RNA from each sample was used to construct the BGI-based mRNA-seq library. In general, the mRNA molecules were purifed using oligo(dT)-attached magnetic beads and then fragmented into small pieces. The cDNA was generated using random hexamer-primed reverse transcription and adaptors were ligated to the ends of these 3′ adenylated cDNA fragments[77]. Then the library was sequenced on BGISEQ platform using 100 bp pair-ended strategy. Eight Gigabase sequencing data for each sample was generated.

## Transcriptomic data analysis

The raw reads were filtered by SOAPfilter, an application included in the SOAPdenovo package(v.2.0)[78]. Then the resulting high quality reads from each sample were mapped against the *Setaria italica* cv. Zhanggu reference genome (v.2.3) (https://ftp.cngb.org/pub/CNSA/data2/CNPhis0000549/Foxtail_millet/) using hisat2 (v.2.0.4)[72][79]. FeatureCounts from the package of the Subread (v.1.6.4)[80] were used to

count the reads that mapped to each one of the protein-coding sequences. Then R-package ballgown (v.2.22.0) was used to calculate the FPKM value of the genes[81]. The differential expression genes were identified using DEseq (v.1.38.0)[82] based on the reads matrix. The differential genes were enriched into pathways using the Fisher's exact test.

## Reporting summary
Further information on research design is available in the Nature Research Reporting Summary linked to this article.

## Data availability
The genomic data of foxtail millet cultivars and RNA-seq data for experimental seedlings generated in this study have been deposited into the NCBI database under accession code PRJNA873890. These data also had been deposited in CNGB Sequence Archive (CNSA) of China National GeneBank DataBase (CNGBdb) with accession code CNP0001521. The *Setaria italica* cv. Zhang gu reference genome (ver. 2.3) used in this study could be found in the CNSA with accession number CNPhis0000549. Other data generated in this study are provided in the Supplementary Data files.

## Code availability
Data and code used for analyses are publicly available at Github[83]. https://zenodo.org/badge/latestdoi/424864991.

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

## Acknowledgements

This research was supported by the Funding of Joint Research on Agricultural Variety Improvement of Henan Province (No. 2022010401, H.Z.), the Major Science and Technology Projects of Yunnan Province (Digitalization, development and application of biotic resource, No. 860 202002AA100007, H.L.), the National Science Foundation (32088102, 31730103, 31825003, E.W.), the Specialty Industry for Key Research and Development Program in Shanxi Academy of Agricultural Sciences (No. YCX2019T01, Z.M.) and Key R&D Program of ShanXi Province (No. 201903D211003, Z.M.). This work was also supported by China National GeneBank (CNGB), Key Laboratory of Genomics, Ministry of Agriculture, BGI-Shenzhen as well as the Danish National Research Foundation (DNRF137, L.G.) for the Center of Microbial Secondary Metabolites.

## Author contributions

H.L., E.W., Y.W. and H.Y. established the concept of the study. Y.W., G.Z., H.Z., J.S., J.W., X.L. and J.W. collected and processed samples. Y.W., S.S., X.W., J.S. and C.J. performed bioinformatics analyses. Y.W., X.W. and E.W. wrote the draft. H.L., S.K.S., L.G., T.W., H.C. revised and edited the manuscript. E.W. conceived the "precision microbial management". All authors have discussed the results, read and approved the contents of the manuscript.

## Competing interests

The authors declare no competing interests.

## Additional information

Yayu Wang [1,9], Xiaolin Wang [2,9], Shuai Sun [1,3], Canzhi Jin [1,4], Jianmu Su [1], Jinpu Wei [1], Xinyue Luo [1,4], Jiawen Wen [1,4], Tong Wei [1], Sunil Kumar Sahu [1], Hongfeng Zou [1], Hongyun Chen [1], Zhixin Mu [5], Gengyun Zhang [1], Xin Liu [1], Xun Xu [1,6], Lone Gram [7], Huanming Yang [1], Ertao Wang [2] ✉ & Huan Liu [1,8] ✉

[1] State Key Laboratory of Agricultural Genomics, BGI-Shenzhen, Shenzhen 518083, China. [2] National Key Laboratory of Plant Molecular Genetics, Chinese Academy of Sciences Center for Excellence in Molecular Plant Sciences, Institute of Plant Physiology and Ecology, Shanghai Institutes for Biological Sciences, Chinese Academy of Sciences, Shanghai 200032, China. [3] BGI-Qingdao, Qingdao 266555, China. [4] College of Life Sciences, University of Chinese Academy of Sciences, Beijing 100049, China. [5] Center for Agricultural Genetic Resources Research, Shanxi Agricultural University, Taiyuan 030031, China. [6] Guangdong Provincial Key Laboratory of Genome Read and Write, BGI-Shenzhen, Shenzhen 518083, China. [7] Department of Biotechnology and Biomedicine, Technical University of Denmark, Søltofts Plads, 2800 Kgs, Lyngby, Denmark. [8] BGI Life Science Joint Research Center, Northeast Forestry University, Harbin 150040, China. [9] These authors contributed equally: Yayu Wang, Xiaolin Wang. ✉ e-mail: etwang@cemps.ac.cn; liuhuan@genomics.cn

