## [Peer Review File · Nature Communications]

Reviewers' Comments:

Reviewer #1:

Remarks to the Author:

For GWAS (or MWAS) multi environmental phenotyping (in at least three environments) is required to estimate the environmental effects, as well as for reproducible and consistent results (marker-trait associations) to be utilized in crop breeding programs. However, in the current manuscript GWAS and MWAS were conducted using single environmental data (Yangling-2013) and may not be sufficient for any conclusive results.

Reviewer #2:

Remarks to the Author:

Wang et al. conducted associations analysis of genotypic, phenotypic, and environmental variables in 827 foxtail millet cultivars by GWAS and MWAS, and identified 257 root microbial biomarkers associated with six key agronomic traits. Additionally, the authors validated the effect of microbial-mediated growth on foxtail millet using marker strains isolated from the field. Also, the authors deciphered the relationship between host variant loci and microbial abundance at the microbial level i.e. how they affect plant growth, and at the genetic level, and extend these biomarker reflections mentioned above to graminaceous plants. I believe this study provides a new way of thinking about how microbial metagenomes, host variant loci affect plant inter-root enrichment and whether these specific loci are conserved in graminaceous plants. Secondly, this study is also informative for the study of inter-rhizosphere mycota of gramineae. The manuscript is well written and this research provides a valuable resource for cereal research and improvement and sheds light on the engineering of high-yielding cultivars in agricultural systems.

Despite the relatively large dataset, I am concerned about a number of issues of the analysis which need to be addressed.

Line 131: The authors used as input data the presence of 70% of OTUs and the phenotypic correlation value < 0.05 and noted that these 70% of OTUs acted as an important component of the community structure. According to the "microbial community structure distribution characterized by a small number of dominant species, a large number of rare species, and a high relative abundance of a few common populations" (Magurran et al., 2004; McGill et al., 2007), these 70% OTUs are an important component of the foxtail millet inter-root community, but they do not necessarily contribute the most to community stability, and it is recommended to increase the assessment of the contribution of these OTUs to community structure.

Line 153: During the authors' evaluation of candidate datasets using linear regression models, the R^2 was low when using only phenotype-related inter-root OTUs, and increased when combining information from SNP loci. What is the contribution of these OTUs associated with SNP loci to community structure? Are they better able to explain the effect of different bacteria on phenotype? Because abundant species tend to be modest in response to community structure and rare species play a more important role in community structure (Yuting Liang et al., 2020; Chao Xiong et al., 2020).

Line 235-237: The authors identified 7 bacterial orders that may be of general interest in the Grass family. It is suggested to focus on the order Verrucomicrobiales, which is related to plant inter-root interactions and may be of interest for inter-root growth in cereals (Eneas Aguirre-von-Wobeser et al., 2018).

Line 449: What is the criteria for splitting the 827 samples into 5 groups at random? Or is this value convenient for later AIC assessment. Also, what is the effect of different groupings, on the AIC value assessment, and is there an optimal threshold for the number in each group?

References:

Magurran et al. (2004). *Measuring Biological Diversity*. Blackwell Publishing, Oxford, UK. ISBN 0-632-05633-9.

McGill et al., (2007). Species abundance distributions: moving beyond single prediction theories to integration within an ecological framework. *Ecology Letters*. 10, 995-1015.2

Yuting Liang, Xian Xiao, Erin E. Nuccio, Mengting Yuan, Na Zhang, Kai Xue, Frederick M. Cohan,

Jizhong Zhou & Bo Sun. (2020) . Differentiation strategies of soil rare and abundant microbial taxa in response to changing climatic regimes. *Environ. Microbiol.* 22, 1327-1340.
Chao Xiong, Ji-Zheng He et al. (2021) . Rare taxa maintain the stability of crop mycobiomes and ecosystem functions. *Environ. Microbiol.* 23, 1907-1924.
Eneas Aguirre-von-wobeser et al. 2018. .Enrichment of Verrucomicrobia, Actinobacteria and Burkholderiales drives selection of bacterial community from soil by maize roots in a traditional milpa agroecosystem. *PLOS.* 13, e0208852.

Reviewer #3:

Remarks to the Author:

Reviewer notes:

In the manuscript "Genome-Wide association studies provide insights into precision genotype-dependent microbial effects in foxtail millet", Wang and colleagues report that plant phenotypic variation in foxtail millet isn't only shaped in a direct manner by plant-genetic variation, but also by indirect effects of plant-genetic variation on rhizosphere microbiota. These microbes then in turn contribute to plant growth promotion (and in some cases root growth suppression). The premise of the manuscript is that missing heritability in plant GWAS is driven in part by the tendency so far in failing to describe the microbes in the plant microbiome. The authors present supporting results that motivate an intriguing model in which plant-genetic variation selects for a host-microbiome that in turn regulates agronomic traits.

Remarkably, the authors show (moderate to strong) effects of marker OTUs in growth promotion and suppression in an allele-specific manner. Such results will be invaluable in directing future microbiome-GWAS related research and in illustrating that the plant microbiota can be used to improve agriculture, albeit in a genotype-dependent manner.

The results arise from large-scale field experiments (phenotypic data) and high-quality genetic data generated by the authors for genome-wide association studies and other statistical genetic studies. Moreover, the authors characterized the rhizosphere/rhizoplane microbial communities of this large mapping panel. All three datasets will be highly valuable for the foxtail community and, more broadly, other plant geneticists.

I enjoyed reading the manuscript, and know that the results from this research will be a solid contribution to the literature. I list a few questions and comments below.

Comments:

I would like to read more about the agronomic importance of millet in the introduction. A previous study, which the present study is partly based on (Jin et al., 2017), could also be mentioned in the introduction or main text already. Information about the (control) lines used, the reference genome and genome size could also be mentioned in the main text, given the strong likelihood that these resources will be heavily used by other authors (and this paper will in turn be cited in part for the data).

Lines 73-74: please describe the pattern/rate of decay in LD here for the reader. One easy summary statistic would be to simply describe the rate at which LD decays to e.g. 80% or 50% of its maximum value. Does it do so within 1 kb? 5 kb? Such information, in the context of the total genome size, would enable one to better understand the SNP density (e.g. 161k SNPs) used in the paper.

Line 146-8: isn't the more interesting (and relevant) question whether (root microbiome + host genotype) explains more than (host genotype)? It isn't clear why the authors instead compare (root microbiome + host genotype) against just (root microbiome), as it isn't typically hypothesized that the root microbiome is the major determinant of host-plant phenotypic variation. The null would in fact be that host-genetics plays the key part in shaping traits; to improve power, one would then add the microbial element to this base model. For a LMM, one approach would be to extend the basic model to consider individual marker OTUs as a fixed effect,

right?

Several of the major conclusions of the paper are based on suggestive P-value thresholds; it's thus unclear how strong the overall observations (and conclusions) are. Please comment.

Minor:

Line 84: "were with" should perhaps be 'exhibited' (?)

Line 84: "with the" should perhaps be 'showed the' (?)

Line 104: what are tolerant genes?

Line 127: "microbiota is" \diamond microbiota are

Line 136: add a percentage sign

Line 153: This sentence ('The best predicting models..') comes very unexpectedly. Please develop/clarify

Line 159: what are the numbers? The number of OTUs? Please clarify

Line 161: It is interesting that the marker OTUs were specific to the traits. On a coarser level (e.g., family or genus), were there taxonomic similarities among the traits?

Line 172: describe positive/negative marker OTUs in more detail

Line 174: HUA12 – is this the reference strain? Please add a few words for researchers that aren't familiar with foxtail

Line 181-3: unclear!

Line 184: What are strains with good performances? Please clarify

Line 198ff.: please spell amino acids, pathways, and other metabolites in a consistent manner

Line 232 and 239: please get rid of the 'both' if it is a list of three

Lines 238-40: the sentence is unclear

Line 274: superscript of reference numbers

Lines 285: This section about the overlap is hard to follow, as the numbers are not explained in detail (which OTUs are which). By the way: "for the 682 of the 219"? Something's wrong here.

Line 331 and 334: remarks about significance should be supported by numbers, please add the stats into the text.

Line 322: "specifically assessed" implies functional assays (that weren't performed here, as far as I can tell). I suggest this section be revised for clarity.

Line 478: How were the OTU abundances normalized?

Line 494: Which type of media was used?

Line 495: universal primers

Line 496: pooled together in equimolar ratios

As an aside: I would consider coming up with a naming convention for the marker OTUs. I understand it is easier to just mention their numbers and OTU ids, but it would benefit the reader if the associated taxonomy was easily accessible in the text or figures. For example, in Figure 2B and C, the bar plots could be ordered in an alphabetical order, or according to classes. This was already done in Figure 6B.

Fig. S2, the Heritability estimates could be colored by phenotype category (e.g. growth versus yield)

Fig. S5 – a subset of the OTUs are highly heritable: Bacilalles and GP4??

Fig. S7 – the Venn diagram labels are unclear. Please check the numbers on the right hand side, as they appear to be in error.

Dear Reviewers,

We greatly appreciate your constructive comments and suggestions that have helped us to significantly improve the manuscript for this revision. We have performed additional analysis that has further validated and strengthened all of our major conclusions in the manuscript. Below we provide comments to each of their suggestions or concerns.

Reviewer #1 (Remarks to the Author):

For GWAS (or MWAS) multi environmental phenotyping (in at least three environments) is required to estimate the environmental effects, as well as for reproducible and consistent results (marker-trait associations) to be utilized in crop breeding programs. However, in the current manuscript GWAS and MWAS were conducted using single environmental data (Yangling-2013) and may not be sufficient for any conclusive results.

RESPONSE: Thanks for raising this point. We do agree with you that the three environmental replicates were used for GWAS research related to traditional agronomic traits, in which the major goal is to dissect the interaction between genotype and phenotype, and eliminate the environmental effects. However, there is currently no method or sample could be followed for large scale MWAS in plant microbiome and agronomic traits. Most of MWAS researches are conducted in the human microbiome to investigate their associations with complex diseases and also highlight the importance of a larger sample size to distinguish the association from background variation(Wang and Jia, 2016). For instance, a large-scale population study including 7,009 individuals from 14 districts within Guangdong province in China reported that regional locations rather than other host or environmental phenotypes were the strongest factor associated with the gut microbiota variations. They further revealed that such regional differences would drastically harmper the MWAS-based microbial models for disease prediction (He et al., 2018).The findings have commonly suggested that the environmental differences have severely reduced the generalizability of the MWAS results. Similarly, it is widely recognized that different soils contain a wide range of bacteria, all of which have a significant impact on the root microbiome, regardless of plant genotype.

In this manuscript, we investigated the interactions among genotype, phenotype and root microbiota. In the same environmental condition, the root sample for each cultivar were harvested from three independent plants and the phenotypic data for each cultivar was collected from ten independent plants. We used a 5-fold cross-validation method to assess the regression model and test its performance and repeated the process for 30 rounds to reduce the noise. Specifically, the original 827 samples were randomly partitioned into 5 equal-sized groups, each group included about 165 samples. We retained 4 groups (662 samples) that were used as training data to train the prediction model, and the remaining single group (165 samples) as the validation data for testing the fitness of the model. The cross-validation process was then repeated 5 times, with each of the 5 sub-group samples used exactly once as

the validation data. Both the sampling and modeling method is a common strategy and widely used in the plant and human gut microbiome researches and provide adequate robustness to data analysis (Bulgarelli et al., 2012; Qin et al., 2012; Rothschild et al., 2018; Zhu et al., 2020). Besides, our data provide insights into precision genotype-dependent microbial effects in foxtail millet. In this case, different genetic locus respond to different microbes under given environmental conditions.

Finally, the conclusions we made were not just based on bioinformatic analysis. We validated the microbial-mediated growth effects on foxtail millet with marker strains (identified by MWAS) isolated from the field. Intriguingly, some of microbial-mediated growth effects on foxtail millet with marker strains isolated from the field can also be reproduced at Shenzhen, but some related effects could not be reproduced at Shenzhen. The effects of marker OTUs in growth promotion and suppression in an allele-specific manner in the plate experiments, indicate a precision genotype-dependent microbial effects in foxtail millet. Our research highlight that plant microbiota can be used to improve agriculture, albeit in a genotype-dependent manner as appreciated by reviewer #2 and reviewer #3.

References:

Bulgarelli, D., Rott, M., Schlaeppi, K., Ver Loren van Themaat, E., Ahmadinejad, N., Assenza, F., Rauf, P., Huettel, B., Reinhardt, R., Schmelzer, E., et al. (2012). Revealing structure and assembly cues for *Arabidopsis* root-inhabiting bacterial microbiota. *Nature* 488, 91-95.

He, Y., Wu, W., Zheng, H.M., Li, P., McDonald, D., Sheng, H.F., Chen, M.X., Chen, Z.H., Ji, G.Y., Zheng, Z.D., et al. (2018). Regional variation limits applications of healthy gut microbiome reference ranges and disease models. *Nature medicine* 24, 1532-1535.

Qin, J., Li, Y., Cai, Z., Li, S., Zhu, J., Zhang, F., Liang, S., Zhang, W., Guan, Y., Shen, D., et al. (2012). A metagenome-wide association study of gut microbiota in type 2 diabetes. *Nature* 490, 55-60.

Rothschild, D., Weissbrod, O., Barkan, E., Kurilshikov, A., Korem, T., Zeevi, D., Costea, P.I., Godneva, A., Kalka, I.N., Bar, N., et al. (2018). Environment dominates over host genetics in shaping human gut microbiota. *Nature* 555, 210-215.

Wang, J., and Jia, H. (2016). Metagenome-wide association studies: fine-mining the microbiome. *Nature reviews Microbiology* 14, 508-522.

Zhu, F., Ju, Y., Wang, W., Wang, Q., Guo, R., Ma, Q., Sun, Q., Fan, Y., Xie, Y., Yang, Z., et al. (2020). Metagenome-wide association of gut microbiome features for schizophrenia. *Nat Commun* 11, 1612.

Reviewer #2 (Remarks to the Author):

Wang et al. conducted associations analysis of genotypic, phenotypic, and environmental variables in 827 foxtail millet cultivars by GWAS and MWAS, and identified 257 root microbial biomarkers associated with six key agronomic traits. Additionally, the authors validated the effect of microbial-mediated growth on foxtail

millet using marker strains isolated from the field. Also, the authors deciphered the relationship between host variant loci and microbial abundance at the microbial level i.e. how they affect plant growth, and at the genetic level, and extend these biomarker reflections mentioned above to graminaceous plants. I believe this study provides a new way of thinking about how microbial metagenomes, host variant loci affect plant inter-root enrichment and whether these specific loci are conserved in graminaceous plants. Secondly, this study is also informative for the study of inter-rhizosphere mycota of gramineae. The manuscript is well written and this research provides a valuable resource for cereal research and improvement and sheds light on the engineering of high-yielding cultivars in agricultural systems.

Despite the relatively large dataset, I am concerned about a number of issues of the analysis which need to be addressed.

Line 131: The authors used as input data the presence of 70% of OTUs and the phenotypic correlation value < 0.05 and noted that these 70% of OTUs acted as an important component of the community structure. According to the "microbial community structure distribution characterized by a small number of dominant species, a large number of rare species, and a high relative abundance of a few common populations" (Magurran et al., 2004; McGill et al., 2007), these 70% OTUs are an important component of the foxtail millet inter-root community, but they do not necessarily contribute the most to community stability, and it is recommended to increase the assessment of the contribution of these OTUs to community structure.

References:

1. Magurran et al. (2004). *Measuring Biological Diversity*. Blackwell Publishing, Oxford, UK. ISBN 0-632-05633-9.
2. McGill et al., (2007). Species abundance distributions: moving beyond single prediction theories to integration within an ecological framework. *Ecology Letters*. 10, 995-1015.2
3. Yuting Liang, Xian Xiao, Erin E. Nuccio, Mengting Yuan, Na Zhang, Kai Xue, Frederick M. Cohan, Jizhong Zhou & Bo Sun. (2020). Differentiation strategies of soil rare and abundant microbial taxa in response to changing climatic regimes. *Environ. Microbiol.* 22, 1327-1340.
4. Chao Xiong, Ji-Zheng He et al. (2021) . Rare taxa maintain the stability of crop mycobiomes and ecosystem functions. *Environ. Microbiol.* 23, 1907-1924
5. Eneas Aguirre-von-wobeser et al. 2018. .Enrichment of Verrucomicrobia, Actinobacteria and Burkholderiales drives selection of bacterial community from soil by maize roots in a traditional milpa agroecosystem. *PLOS*. 13, e0208852.

RESPONSE: Thanks for the detailed suggestions. To ensure most of the samples could be used in the process of model construction, we use the 1,004 common OTUs

(with a 70% occurrence in all samples) as the input data as these OTUs commonly exist in the root zone of foxtail millet cultivars. These 1,004 common OTUs (here called common sub-community) occupied an average of 61.30% of total abundance in the whole community.

First, the alpha diversity of common sub-community (1,004 OTUs) and whole community was calculated, respectively. The common sub-community showed higher evenness and lower diversity than the whole community (14,689 OTUs) (Figure S4A and B).

Next, we defined abundant or rare OTUs according to the previously suggested rules (Liang et al., 2020; Xiong et al., 2021). Briefly, the OTUs with relative abundances $\geq 0.1\%$ across more than 50% samples were defined as abundant OTUs (AT), whereas the OTUs with relative abundances $< 0.01\%$ across more than 30% samples but never abundant ($\geq 0.1\%$) more than 30% samples were defined as rare OTUs (RT). Those OTUs neither belonging to abundant taxa nor rare taxa were defined as moderate OTUs (MT). The OTU distribution showed that the rare abundant OTUs (RT) have most of the OTU numbers (93.38%) in the whole community, but covered an average of 39.26% of total abundance, similar to a previous study (Liang et al., 2020). In the common sub-community, the moderate OTUs (covered 83.67% of OTU numbers) were abundant, followed by abundant OTUs (12.85%) and rare OTUs (3.48%) (Tables S5). We further performed network analysis to disentangle the ecological role and co-occurrence patterns of 1004 OTUs in the common sub-community using R packages igraph. Abundant OTUs (AT) had significantly higher values of the degree, closeness, betweenness centrality and hub scores than both rare and moderate OTUs in the network (Figure S4E, Kruskal-Wallis test with $P < 0.001$). It means abundant OTUs play important role in sustaining the stability of the microbial community.

Moreover, we evaluated the correlations of the whole microbial community and common sub-community and marker OTU sub-communities to six growth and yield traits of foxtail millet using the mantel test. The common sub-community showed more significant correlations with growth trait MSW and MSPD than the whole community (Figure S4C). The marker OTU communities generated from the common sub-communities showed more significant correlations with three growth traits (TSLW, MSPD, MSW) and two yield traits (MSPD and PGW) than whole and common sub-community, respectively (Figure S4C).

Furthermore, we added the assessment of the community stability of common OTUs (with 0.7 occurrence frequency) using the average variation degree (AVD) index, which is used as an indicator of microbiome stability and to evaluate soil microbiome stability (Xun et al., 2021). AVD is calculated using the deviation degree from the mean of the normally distributed OTU relative abundance using the following equation. A lower AVD value indicates higher microbiome stability.

$$AVD = \frac{\sum_{i=1}^n \frac{|x_i - \bar{x}_i|}{\delta_i}}{k \times n}$$

(x_i is the rarefied abundance of the OTU in one sample, \bar{x}_i is the average rarefied abundance of the OTU in one sample group, and δ_i is the standard. k is the number of samples in one sample group, n is the number of OTUs in each sample group.)

To compare the structure stability with the common OTU community, we extracted the OTU sub-tables with 0.3 and 0.5 occurrence frequency from the whole communities, respectively. The AVD value (0.812) from the common OTU community here (called common sub-community) is slightly lower than the AVDs from 0.3 and 0.5-sub community (Figure S4D), suggesting higher microbiome stability of common sub-community than 0.3 and 0.5 sub-microbial community.

Collectively, the common OTUs dominated the whole community, contributed to higher stability of the microbial community and provide important components to select the beneficial microbes to the growth and yield of foxtail millet. We also added description as described below in the result section from lines 150 to 167, and provided Figure S4 and Table S5 in supplementary materials. We also introduced the analysis content with subtitles “Definition of abundant and rare microbial OTUs”, “The AVD value calculation”, and “The microbial co-occurrence network analysis” in the method section from lines 544 to 573 accordingly.

In the result section:

“The 1,004 rhizoplane operational taxonomic units (OTUs) with a 70% occurrence in all samples (here defined as common OTUs), covering an average of 61.30% of total abundances were used as the input data as these OTUs commonly exist in the root zone of foxtail millet cultivars. The common sub-community (1,004 common OTUs) showed higher evenness and correlations with the growth traits than the whole microbial community (Figure S4 A-C). The average variation degree (AVD) index from the common sub-community, 0.5 and 0.3 sub-community (OTUs with 50% and 30% occurrence), were calculated to assess the microbiome stability. The common sub-community had a lower AVD value than the other two sub-communities, indicating that it has a more stable microbiome (Figure S4 D). In the common sub-communities, the moderate OTUs (covered 83.67% of OTU numbers) were abundant, followed by abundant OTUs (12.85%) and rare OTUs (3.48%) (Table S5). The network analysis was used to disentangle the ecological role and co-occurrence patterns of 1,004 OTUs in the common sub-community. Abundant OTUs (ATs) had significantly higher values of the degree, closeness, betweenness centrality and hub scores than both rare (RTs) and moderate OTUs (MTs) in the network (Figure S4E, Kruskal-Wallis test with $P < 0.001$), indicating their important roles in sustaining the stability of the microbial community. Thus, the candidate OTUs that were significantly correlated with the traits (adjust $P < 0.05$) were selected from the common sub-community and used as the input of the predicting models (Table S6).”

Figure S4 The structure of a whole microbial community and common sub-community in the rhizoplane of foxtail millet. The violin chart display the alpha diversities of the whole community and common sub-community in rhizoplane, including evenness (A) and Shannon index(B). The common sub-community had higher evenness but less diversity than the whole microbial community. The significant difference was determined by the Kruskal-Wallis test. (C) the correlations of different microbial communities (whole, common and marker) to each trait were detected by Mantel tests. (D) the barplot display the average variation degree (AVD) value of different microbial communities, such as 0.3 sub-community, 0.5 sub-community, common sub community and whole community. (E) topological features (degree, closeness, betweenness centrality and hub scores) of the co-occurrence network were compared among abundant OTUs (ATs), moderate OTUs (MTs) and rare OTUs (RTs) of common sub-community.

References:

Liang, Y., Xiao, X., Nuccio, E.E., Yuan, M., Zhang, N., Xue, K., Cohan, F.M., Zhou, J., and Sun, B. (2020). Differentiation strategies of soil rare and abundant microbial taxa in response to changing climatic regimes. *Environmental Microbiology* 22, 1327-1340.

Xiong, C., He, J.Z., Singh, B.K., Zhu, Y.G., Wang, J.T., Li, P.P., Zhang, Q.B., Han, L.L., Shen, J.P., and Ge, A.H. (2021). Rare taxa maintain the stability of crop microbiomes and ecosystem functions. *Environmental Microbiology* 23, 1907-1924.

Xun, W., Liu, Y., Li, W., Ren, Y., Xiong, W., Xu, Z., Zhang, N., Miao, Y., Shen, Q., and Zhang, R. (2021). Specialized metabolic functions of keystone taxa sustain soil microbiome stability.

Line 153: During the authors' evaluation of candidate datasets using linear regression models, the R2 was low when using only phenotype-related inter-root OTUs, and

increased when combining information from SNP loci. What is the contribution of these OTUs associated with SNP loci to community structure? Are they better able to explain the effect of different bacteria on phenotype? Because abundant species tend to be modest in response to community structure and rare species play a more important role in community structure (Yuting Liang et al., 2020; Chao Xiong et al., 2020).

RESPONSE: Thanks for your suggestions. Among 257 marker OTUs, 219 OTUs were associated with the host genetic variations, here called SNP-associated marker OTUs and the remaining 38 OTUs not associated with the genetic variations, were called non SNP-associated marker OTUs. The topology characteristics of the network from 257 marker OTUs were calculated to quantitatively describe the species' role in communities using R packages igraph. The degree, closeness, betweenness centrality and hub score had no differences between the SNP-associated marker OTUs and non SNP-associated marker OTUs (Figure S5A). It means SNP- and non SNP-associated marker microbes had a similar contribution to community stability. Moreover, the abundant marker OTUs (AMTs) showed a significantly higher value of the degree, closeness, and betweenness centrality than both rare (RMTs) and moderate OTUs (MMTs) in the network (Figure S5B), indicating the abundant marker OTUs (AMTs) have more important roles in community structure.

We also calculated the correlations of the SNP-associated marker OTUs and non SNP-associated marker OTUs with growth and yield traits, separately (Table S11). SNP-associated marker OTUs showed higher correlations with the growth trait (MSPD and MSW) than non SNP-associated marker OTUs, while having no difference in correlations with trait TSLW, MSPW, PGW and MSPL (Table S11). It means that the genotype-dependent marker OTUs might explain more variances in plant growth traits.

We also updated the following content in the result section from line 190 to 194 and line 342 to 345. We added the Figure S5 and Table S11 in supplementary materials accordingly.

In the result section:

Line 190 to 194 “Network analysis of 257 marker OTUs showed that the abundant marker OTUs (AMTs) had a significantly higher value of the degree, closeness and betweenness centrality than both rare (RMTs) and moderate marker OTUs (MMTs), indicating the abundant marker OTUs have more important roles in community structure (Table S4 and Figure S5, Kruskal-Wallis test with $P<0.05$).”

Line 342 to 345 “Moreover, Mantel test also showed that SNP-associated marker OTUs had higher correlations with the growth trait (MSPD and MSW) than non SNP-associated marker OTUs, while having no difference in correlations with trait TSLW, MSPW, PGW and MSPL (Table S11). It means that the genotype-dependent marker OTUs might explain more variances in plant growth traits.”

Line 235-237: The authors identified 7 bacterial orders that may be of general

interest in the Grass family. It is suggested to focus on the order Verrucomicrobiales, which is related to plant inter-root interactions and may be of interest for inter-root growth in cereals(Eneas Aguirre-von-Wobeser et al., 2018).

RESPONSE: Thanks for your suggestions. We noticed that only 23 OTUs from the root microbiome of foxtail millet (a total of 14869 OTUs) were annotated to the *Verrucomicrobiales* order, covering four genera such as *Luteolibacter*, *Verrucomicrobium*, *Roseimicrobium* and *Prostheco bacter*. Only one OUT (*Luteolibacter* OTU_109) from the common OTU set showed lower heritability and was not correlated with six growth and yield traits. Although members of Verrucomicrobia and other phyla found in the rhizosphere may establish beneficial plant-microbe interactions with maize roots in milpas, but it is not common in the foxtail millet root zone. We also check the association between the *Luteolibacter* OTU_109 and genetic SNPs and found that the SNP si7:18015612 was significantly associated with this OTU. The gene around the SNP locus encoded a protein acetyl-coenzyme A synthetase, which catalyzes the production of acetyl-CoA, an activated form of acetate that can be used for lipid synthesis or energy generation. This information might be useful in future research in plant and *Verrucomicrobiales* interaction.

Line 449: What is the criteria for splitting the 827 samples into 5 groups at random? Or is this value convenient for later AIC assessment. Also, what is the effect of different groupings, on the AIC value assessment, and is there an optimal threshold for the number in each group?

RESPONSE: Thanks for your comment. We didn't describe the process clearly in the previous version. We did five-fold cross-validation, which is a common strategy and widely used to improve the model performance. The original 827 samples were randomly partitioned into 5 equal-sized groups, each group included about 165 samples. We retained 4 groups (662 samples) that were used as training data to train the prediction model, and the remaining single group (165 samples) as the validation data for testing the fitness of the model. The cross-validation process was then repeated 5 times, with each of the 5 sub-group samples used exactly once as the validation data. The prediction model with the lowest AIC value was selected in each training data set. We didn't set a threshold for each group. The fitness of the prediction model was then estimated according to correlation efficiency (R^2) of the observation and prediction of the trait in the testing data set. The predictive models generated from different sample groups show different powers in the prediction of the test data, so we repeated the five-fold cross-validation process for 30 rounds. Finally, the best prediction model with the highest R^2 for the trait was retained. We listed the R^2 of observation and prediction value of each trait from 30 rounds in Figure S3. We also added the description in the method section (lines 504 to 515).

Reviewer #3 (Remarks to the Author):

Reviewer notes:

In the manuscript “Genome-Wide association studies provide insights into precision genotype-dependent microbial effects in foxtail millet”, Wang and colleagues report that plant phenotypic variation in foxtail millet isn’t only shaped in a direct manner by plant-genetic variation, but also by indirect effects of plant-genetic variation on rhizosphere microbiota. These microbes then in turn contribute to plant growth promotion (and in some cases root growth suppression). The premise of the manuscript is that missing heritability in plant GWAS is driven in part by the tendency so far in failing to describe the microbes in the plant microbiome. The authors present supporting results that motivate an intriguing model in which plant-genetic variation selects for a host-microbiome that in turn regulates agronomic traits.

Remarkably, the authors show (moderate to strong) effects of marker OTUs in growth promotion and suppression in an allele-specific manner. Such results will be invaluable in directing future microbiome-GWAS related research and in illustrating that the plant microbiota can be used to improve agriculture, albeit in a genotype-dependent manner.

The results arise from large-scale field experiments (phenotypic data) and high-quality genetic data generated by the authors for genome-wide association studies and other statistical genetic studies. Moreover, the authors characterized the rhizosphere/rhizoplane microbial communities of this large mapping panel. All three datasets will be highly valuable for the foxtail community and, more broadly, other plant geneticists.

I enjoyed reading the manuscript, and know that the results from this research will be a solid contribution to the literature. I list a few questions and comments below.

Comments:

I would like to read more about the agronomic importance of millet in the introduction. A previous study, which the present study is partly based on (Jin et al., 2017), could also be mentioned in the introduction or main text already. Information about the (control) lines used, the reference genome and genome size could also be mentioned in the main text, given the strong likelihood that these resources will be heavily used by other authors (and this paper will in turn be cited in part for the data).

RESPONSE: Thanks for your suggestions. We have added the description of the agronomic importance of foxtail millet in the introduction section from lines 58 to 65.

We also mentioned the main result of root microbiome in our previous study in lines 69 and 72 in the introduction section. We have added the information of Zhanggu reference genome, which is ~423Mb in the main text in line 82.

Lines 73-74: please describe the pattern/rate of decay in LD here for the reader. One easy summary statistic would be to simply describe the rate at which LD decays to e.g.

80% or 50% of its maximum value. Does it do so within 1 kb? 5 kb? Such information, in the context of the total genome size, would enable one to better understand the SNP density (e.g. 161k SNPs) used in the paper.

RESPONSE: We calculated the rate of LD decay based on the detected 161,562 SNPs as per your suggestion, and found that the LD decays to $r^2 =$ half at the distance of 9 kb. Considering the genome size of 400 Mb, the average distance between neighboring SNPs is 2.5 kb. It roughly means more than 3 SNPs for each linkage block, and our SNP dataset should be dense enough to perform genome-wide association analysis. We have added a plot of genome-wide linkage disequilibrium (LD) decay as Figure S1B and revised the description in line 86 as follows:

“The SNPs were evenly distributed along chromosomes and linkage disequilibrium (LD) decays to $r^2 =$ half at the distance of 9 kb (Figure S1A and B)”

Figure S1B Plot of genome-wide linkage disequilibrium (LD) decay. x-axis represents the genetic distance (Kb) between SNPs and the y-axis represents LD value (r^2). Horizontal and vertical lines represent half LD and LD decay distance respectively, and LD decays to its half decay at the distance at 9 kb.

Line 146-8: isn't the more interesting (and relevant) question whether (root microbiome + host genotype) explains more than (host genotype)? It isn't clear why the authors instead compare (root microbiome + host genotype) against just (root microbiome), as it isn't typically hypothesized that the root microbiome is the major determinant of host-plant phenotypic variation. The null would in fact be that host-genetics plays the key part in shaping traits; to improve power, one would then add the microbial element to this base model. For a LMM, one approach would be to extend the basic model to consider individual marker OTUs as a fixed effect, right?

RESPONSE: Yes, thanks for your helpful suggestions. The null hypothesis should be that host-genetics plays a key part in shaping traits; to improve power, one would then add the microbial element to this base model. Actually, we had performed linear

regression analysis according to three steps. Firstly, we constructed the linear regression model using candidate SNPs as variables alone; Secondly, we constructed the linear regression model using candidate OTUs as variables alone; and finally, we added the effective SNPs in the model to stepwise regress against the phenotype, then the microbial OTUs were added, followed by interactions of SNPs and OTUs. To reduce the noise in the estimated model performance, we repeated the five-fold cross-validation process 30 times and reported the mean performance across all folds and all repeats. The variation of the trait predicted by genetic SNPs alone (Green dots), rhizoplane OTUs alone (blue dots) or by host genetic SNPs and rhizoplane OTUs combined (orange dots) are displayed in Figure S3, respectively. We also revised the descriptions as follows in the result and method section, correspondingly.

In the result section:

Line 139 to 146: “To explore the contributions of genetic variations to plant performance, linear regression models were used to calculate the role of host genotypes on key growth (TSLW, MSPD, MSW)- and yield (MSPW, PGW, MSPL)-related traits of the 827 different foxtail millet cultivars. Considering no SNPs associations with phenotypes TSLW and MSPW under suggestive thresholds, we extended the candidate SNPs ($P < 1.0 \times 10^{-4}$) as inputs of the linear regression models^{34,35} (Table S4). After performing thirty rounds of five-fold cross-validation, the genetic SNP markers in predicting model could explain an average of 32.82%, 28.55%, 47.27%, 15.02%, 38.89% and 64.60% of the variances in TSLW, MSPD, MSW, MSPW, PGW and MSPL in the testing data, respectively (Figure S3).”

Line 180 to 186: “The correlation coefficients only using genotype as variables were obviously higher than that only using root microbiota as variables in several agronomic traits such as MSW and MSPL. However, in the traits MSPD and MSPW, the contribution of root microbiota to phenotypic plasticity was higher when root microbiota variables were used instead of genotype variables alone, indicating a different contribution of host genotype and root microbiota to phenotypic plasticity. The combination of host genotype and root microbiota significantly promoted the explanation of variations in all six traits than genotype and root microbiota alone (Wilcoxon rank test, $P < 0.001$) except for the trait MSPL (Figure S3).”

Figure S3 The R^2 between the observations and predictions of traits in the testing dataset from thirty different repeats of five-fold cross validation. The variation of the trait was predicted by genetic SNPs alone (Green dots), rhizoplane OTUs alone (blue dots) or by host genetic SNPs and rhizoplane OTUs combined (orange dots), respectively.

Several of the major conclusions of the paper are based on suggestive P-value thresholds; it's thus unclear how strong the overall observations (and conclusions) are. Please comment.

RESPONSE: The suggestive ($1/N$) P-value thresholds were set to control the genome-wide type 1 error rate (Duggal et al., 2008), and N represented the effective number of independent SNPs used in the genome-wide association study (GWAS). So we defined the suggestive P-value with $1/49,512=2.01e-5$. The suggestive P-value threshold was first proposed by Lander and Kruglyak, and represents the statistical evidence that would be expected to occur one false positive in a genome scan under the null hypothesis (Lander & Kruglyak, 1995). The suggestive association threshold was commonly used in the GWAS of humans, animals and plants to identify many potential SNPs for consideration of association (Li et al., 2013; Reed et al., 2015, Guo et al., 2017; Yang et al., 2015; Arvanitis et al., 2020). The detected genes or loci above statistical significance was not the only condition to determine the conclusion, and thus some evidence to support the conclusion was also given, for example, the gene functions, and plant growth promotion assay. Taken together, the overall findings of our investigation are convincing.

References:

Arvanitis, M., Tampakakis, E., Zhang, Y., Wang, W., Auton, A., 23andMe Research Team, Dutta, D., Glavaris, S., Keramati, A., Chatterjee, N., Chi, N. C., Ren, B., Post, W. S., & Battle, A. (2020). Genome-wide association and multi-omic analyses reveal ACTN2 as a gene linked to heart failure. *Nature communications*, *11*(1), 1122. <https://doi.org/10.1038/s41467-020-14843-7>

Duggal, P., Gillanders, E. M., Holmes, T. N., & Bailey-Wilson, J. E. (2008). Establishing an adjusted p-value threshold to control the family-wide type 1 error in genome wide association studies. *BMC genomics*, *9*, 516. <https://doi.org/10.1186/1471-2164-9-516>

Lander, E., & Kruglyak, L. (1995). Genetic dissection of complex traits: guidelines for interpreting and reporting linkage results. *Nature genetics*, *11*(3), 241–247. <https://doi.org/10.1038/ng1195-241>

Li, H., Peng, Z., Yang, X., Wang, W., Fu, J., Wang, J., Han, Y., Chai, Y., Guo, T., Yang, N., Liu, J., Warburton, M. L., Cheng, Y., Hao, X., Zhang, P., Zhao, J., Liu, Y., Wang, G., Li, J., & Yan, J. (2013). Genome-wide association study dissects the genetic architecture of oil biosynthesis in maize kernels. *Nature genetics*, *45*(1), 43–50. <https://doi.org/10.1038/ng.2484>

Guo, Y., Huang, Y., Hou, L., Ma, J., Chen, C., Ai, H., ... & Ren, J. (2017). Genome-wide detection of genetic markers associated with growth and fatness in four pig populations using four approaches. *Genetics Selection Evolution*, *49*(1), 21.

Reed, E., Nunez, S., Kulp, D., Qian, J., Reilly, M. P., & Foulkes, A. S. (2015). A guide to genome-wide association analysis and post-analytic interrogation. *Statistics in medicine*, *34*(28), 3769–3792. <https://doi.org/10.1002/sim.6605>

Yang, W., Guo, Z., Huang, C., Wang, K., Jiang, N., Feng, H., Chen, G., Liu, Q., & Xiong, L. (2015). Genome-wide association study of rice (*Oryza sativa* L.) leaf traits with a high-throughput leaf scorer. *Journal of experimental botany*, *66*(18), 5605–5615. <https://doi.org/10.1093/jxb/erv100>

Minor:

Line 84: “were with” should perhaps be 'exhibited' (?)

RESPONSE: Thanks. We revised “were with” as “exhibited”.

Line 84: “with the” should perhaps be 'showed the' (?)

RESPONSE: Thanks. We revised “with the” as “showed the”.

Line 104: what are tolerant genes?

RESPONSE: We have distinguished the drought stress-responsive genes and tolerant genes in the main text. We revised the content as follows:

“Secondly, numerous drought stress-responsive (*PP2C* (Shi et al., 2022), *ARR12* (Nguyen et al., 2016), *NPF1.2*(Parmar et al., 2019), *NPF4.6* (Kuromori et al., 2018), *WDR26* (Chuang et al., 2015), *Plastocyanin-like protein* (Huang et al., 2008), *CPK2a* (Cieřla et al., 2016), *PIP5K1* (Mikami et al., 1998)) and tolerant genes (*APX* (Rossel et al., 2006), *DTX12* (Lu et al., 2019), *bHLH35* (Dong et al., 2014; Jiang et al.,

2019), *Thioredoxin fold domain containing protein* (Jimenez et al., 2012), *SAPK9* (Li et al., 2021), *Ca²⁺-transporting ATPase* (Yang et al., 2018), *InsP3* (Khodakovskaya et al., 2010), *E3 ubiquitin-protein ligase MIEL1* (Lee and Seo, 2016)) were found to be strongly associated with the growth and yield traits (Table S3).”

References:

Shi, Y., Liu, X., Zhao, S. & Guo, Y. The PYR- PP2C- CKL2 module regulates ABA- mediated actin reorganization during stomatal closure. *New Phytologist* 233, 2168-2184 (2022).

Nguyen, K. H. et al. Arabidopsis type B cytokinin response regulators ARR1, ARR10, and ARR12 negatively regulate plant responses to drought. *Proceedings of the National Academy of Sciences* 113, 3090-3095 (2016).

Parmar, R. et al. Transcriptional profiling of contrasting genotypes revealed key candidates and nucleotide variations for drought dissection in *Camellia sinensis* (L.) O. Kuntze. *Scientific reports* 9, 1-12 (2019).

Kuromori, T., Seo, M. & Shinozaki, K. ABA transport and plant water stress responses. *Trends in plant science* 23, 513-522 (2018).

Chuang, H.-w., Feng, J.-H., Feng, Y.-L. & Wei, M.-J. An Arabidopsis WDR protein coordinates cellular networks involved in light, stress response and hormone signals. *Plant Science* 241, 23-31 (2015).

Huang, D., Wu, W., Abrams, S. R. & Cutler, A. J. The relationship of drought-related gene expression in *Arabidopsis thaliana* to hormonal and environmental factors. *Journal of experimental Botany* 59, 2991-3007 (2008).

Cieřła, A. et al. A role for barley calcium-dependent protein kinase CPK2a in the response to drought. *Frontiers in plant science* 7, 1550 (2016).

Mikami, K., Katagiri, T., Iuchi, S., Yamaguchi, Shinozaki, K. & Shinozaki, K. A gene encoding phosphatidylinositol- 4- phosphate 5- kinase is induced by water stress and abscisic acid in *Arabidopsis thaliana*. *The Plant Journal* 15, 563-568 (1998).

Rossel, J. B. et al. A mutation affecting ASCORBATE PEROXIDASE 2 gene expression reveals a link between responses to high light and drought tolerance. *Plant, Cell & Environment* 29, 269-281 (2006).

Lu, P. et al. Overexpression of cotton a DTX/MATE gene enhances drought, salt, and cold stress tolerance in transgenic *Arabidopsis*. *Frontiers in plant science* 10, 299 (2019).

Dong, Y. et al. A novel bHLH transcription factor PebHLH35 from *Populus euphratica* confers drought tolerance through regulating stomatal development, photosynthesis and growth in *Arabidopsis*. *Biochemical and biophysical research communications* 450, 453-458 (2014).

Jiang, L. et al. The AabHLH35 transcription factor identified from *Anthurium andraeanum* is involved in cold and drought tolerance. *Plants* 8, 216 (2019).

Jimenez, A. J., Castillo, A. G., Valpuesta, V., Borsani, O. & Botella, M. A. The Arabidopsis TETRATRICOPEPTIDE THIOREDOXIN-LIKE Gene Family Is Required for. *Plant Physiol* 158, 1253 (2012).

Li, X. et al. OsMADS23 phosphorylated by SAPK9 confers drought and salt tolerance by regulating ABA biosynthesis in rice. *PLoS genetics* 17, e1009699 (2021).

Yang, Y. et al. Involvement of an ABI-like protein and a Ca²⁺-ATPase in drought tolerance as revealed by transcript profiling of a sweetpotato somatic hybrid and its parents *Ipomoea batatas* (L.) Lam. and *I. triloba* L. *PLoS one* 13, e0193193 (2018).

Khodakovskaya, M. et al. Increasing inositol (1, 4, 5)- trisphosphate metabolism affects drought tolerance, carbohydrate metabolism and phosphate- sensitive biomass increases in tomato. *Plant biotechnology journal* 8, 170-183 (2010).

Lee, H. G. & Seo, P. J. The Arabidopsis MIEL1 E3 ligase negatively regulates ABA signalling by promoting protein turnover of MYB96. *Nature Communications* 7, 1-11 (2016).

Beasley, T. M., Erickson, S. & Allison, D. B. Rank-based inverse normal transformations are increasingly used, but are they merited? *Behavior genetics* 39, 580-595 (2009).

Line 127: “microbiota is” ◊ microbiota are

RESPONSE: We corrected it.

Line 136: add a percentage sign

RESPONSE: We have added the percentage sign in the content.

Line 153: This sentence (‘The best predicting models.’) comes very unexpectedly. Please develop/clarify

RESPONSE: We revised the sentence as follows:

“The predictive models with best prediction accuracy for the phenotypes using the SNP and OTU variables were retained, that explained 53.42%, 63.73%, 70.54%, 50.16%, 55.88%, and 54.82% variations for TSLW, MSPD, MSW, MSPW, PGW and MSPL trait, respectively, resulting in a final set of 257 marker OTUs (Figure 2 A-F, Table S6)”

Line 159: what are the numbers? The number of OTUs? Please clarify

RESPONSE: Yes, they are OTU numbers in each phylum. We have revised this sentence as follows:

“The top five abundant phyla were *Proteobacteria* (with 68 OTUs), *Actinobacteria* (54 OTUs), *Bacteroidetes* (36 OTUs), *Acidobacteria* (35 OTUs), and *Firmicutes* (33 OTUs) (Figure 3A)”

Line 161: It is interesting that the marker OTUs were specific to the traits. On a

coarser level (e.g., family or genus), were there taxonomic similarities among the traits?

RESPONSE: We listed the composition comparisons of the marker OTUs (Figure S6A), genera (Figure S6B) and families (Figure S6C) among six traits here. At the family level, only one family *Chitinophagaceae* was shared by all six traits, while no OTUs and genus were shared by all six traits.

Figure S6 The Venn diagram displaying the overlap of microbial markers among six traits at OTUs level (A), genus level (B) and family level (C), respectively. Only one family *Chitinophagaceae* was shared by all the six traits, while neither OTU nor genus was shared by all six traits.

Line 172: describe positive/negative marker OTUs in more detail

RESPONSE: We have added the marker OTU information to the content in lines 213 to 217 as follows:

“Representative cultivated strains of six positive marker OTUs (*Acid550* to *Acidovorax* OTU_46, *Baci299* to *Bacillaceae* OTU_22228, *Kita594* to *Kitasatospora* OTU_8, *Baci154* to *Bacillus* OTU_19414, *Baci312* to *Bacillus* OTU_25704 and *Baci429* to *Bacillales* OTU_381) and four negative marker OTUs (*Shin228* to *Shinella* OTU_37, *Baci81* to *Bacillus* OTU_54, *Baci173* to *Bacillaceae* OTU_19835 and *Baci554* to *Bacillaceae* OTU_28133) with top beta estimation in the regression model were selected for the validation experiments (Figure 3A and Figure S7).”

Line 174: HUA12 – is this the reference strain? Please add a few words for researchers that aren’t familiar with foxtail

RESPONSE: We have revised the sentence and added the description to the cultivar in line 219:

“We co-cultivated these 10 biomarker strains with foxtail millet Huagu12 (a inbred cultivar of foxtail millet (*Setaria italica*) at Shenzhen, China) for 7-days in sterilized

plates, and observed altered root lengths and plant heights compared with the control (Figure 3B and Figure S7A).”

We have also added the introduction to Huagu12 in the method section (line 635) like this:

“Huagu12 is a foxtail millet (*Setaria italica*) cultivar bred by BGI Institute of Applied Agriculture (Shenzhen, China).”

Line 181-3: unclear!

RESPONSE: In the plate experiment, two negative marker strains (Shin228 respond to *Shinella* OTU_37, and Baci81 respond to *Bacillus* OTU_54) didn't show a suppressive effect on foxtail millet growth but displayed growth promoting effect. We hypothesized two likely possibilities: first, the strain *Shin228* and *Baci81* was wrongly identified as the target strains of marker OTUs. This error would have occurred since we identified the representative strain of marker OTUs based on 16S rDNA sequence similarity (cutoff $\geq 97\%$); second, the suppressive effects of these strains might function only under the cooperation with other specific strains in the root microbiome. We have revised this sentence in lines 226 to 230 as described below:

“While the negative marker strains *Shin228* (*Shinella* OTU 37) and *Baci81* (*Bacillus* OTU 54) exhibited growth-promoting effects, they may only function in special root microbial flora in collaboration with other strains or be mistakenly identified as representative strains due to high 16S rDNA sequence similarities with negative marker OTU 37 and 54.”

Line 184: What are strains with good performances? Please clarify

RESPONSE: We added the strain information in the content and revised the sentence in lines 231 to 234 as follows:

“Next, by watering millet seedlings grown in sterilized soil with the bacterial suspensions, we validated the effects of four positive marker strains (Kita594, Baci299, Baci154, and Acid550) and two negative marker strains (Baci173 and Baci554) with good promoting or suppressing performances on plant growth in a plate experiment.”

Line 198ff.: please spell amino acids, pathways, and other metabolites in a consistent manner

RESPONSE: Thanks. We consistently spell the pathways with the plot. We revised the sentence in lines 247 to 252 as follows:

“For example, the differentially expressed genes caused by growth-promoting strains were mainly enriched in the pathways such as Phenylalanine, tyrosine and tryptophan biosynthesis (ko00400), Biosynthesis of amino acids (ko01230), Phenylalanine

metabolism (ko00360), Carbon fixation in photosynthetic organisms (ko00710), Photosynthesis-antenna proteins (ko00196), Photosynthesis (ko00195), MAPK signaling pathway-plant (ko04016), Plant-pathogen interaction (ko04626), Diterpenoid biosynthesis (ko00904), Monoterpenoid biosynthesis (ko00902), alpha-Linolenic acid metabolism (ko00592) and Selenocompound metabolism(ko00450).”

Line 232 and 239: please get rid of the ‘both’ if it is a list of three

RESPONSE: Thanks. We deleted the word “both” in the sentence.

Lines 238-40: the sentence is unclear

RESPONSE: It means that the abundances of microorganisms from the order *Bacillales*, *Actinomycetales*, *Burkholderiales*, *Rhizobiales*, *Myxococcales*, *Sphingobacteriales* were more easily impacted by genetic variations in sorghum, maize and foxtail millet. I revised this sentence in line 299 as follows:

“These results hence indicated that the microorganisms in these bacterial orders were more sensitive to genetic variations across both sorghum, maize and foxtail millet.”

Line 274: superscript of reference numbers

RESPONSE: Thanks. We have checked the superscript of all reference numbers.

Lines 285: This section about the overlap is hard to follow, as the numbers are not explained in detail (which OTUs are which). By the way: “for the 682 of the 219”? Something’s wrong here.

RESPONSE: Thanks for the nice comment. We have revised the content from line 343 to 345 as follows:

“We discovered that 219 of the SNP-associated OTUs overlapped with the marker OTUs in our data sets, covering 85.2% of 257 marker OTUs (Figure S10A, 219 out of 257 = 85.2%). 682 SNP loci were significantly associated with 219 marker OTUs (here called marker OTU-associated SNPs). However, for the 682 marker OTU-associated SNPs, only 4 overlapped with the 45 non-redundant marker SNPs (GWAS identified) that were associated with the aforementioned agronomic traits of foxtail millet (Figure S10B).”

Line 331 and 334: remarks about significance should be supported by numbers, please add the stats into the text.

RESPONSE: Thanks. We have added the significance in lines 397 and 400 as described below.

“Intriguingly, we found that strain 594 had a statistically significantly shoot-promoting effect only on the allele cultivars, but not on reference cultivars (Figure 6C-E, adjusted $P < 0.05$ by ANOVA-LSD), supporting that plant-growth promoting rhizobacteria support genotype-dependent cooperation with the plant. Consistently, we observed strong root growth inhibition in seedlings inoculated with the growth-suppressing strain 173 (Figure 6F-H, adjusted $P < 0.05$ by ANOVA-LSD), and a more significant suppressing effect on root length was observed in the allele cultivars (C1296 and C1021) compared to the reference cultivars (C946 and C306).”

Line 322: “specifically assessed” implies functional assays (that weren’t performed here, as far as I can tell). I suggest this section be revised for clarity.

RESPONSE: Yes, Thanks. We have revised the sentence in line 387 as follows:

“Finally, based on cultivars with different genotypes, the influence of functional SNPs on marker OTU abundance was thoroughly examined.”

Line 478: How were the OTU abundances normalized?

RESPONSE: We applied Rank-Based Inverse Normal Transformations (Rank-based INT) to normalize the OTU abundance, which is a way of transforming the sample distribution of a continuous variable to make it appear more normally distributed and commonly applied during GWAS of nonnormally distributed traits (Beasley et al., 2009). Rank-based INT entails creating a modified rank variable and then computing a new transformed value of the phenotype for the i th subject. The following is a detailed description of the calculation formula:

$$y_i^t = \Phi^{-1} \left(\frac{r^i - c}{N - 2c + 1} \right)$$

where r_i is the ordinary rank of the i th case among the N observations and Φ^{-1} denotes the standard normal quantile (or probit) function. The value of $c = 3/8$ is recommended by Blom (Blom et al., 1958).

We revised the sentence and added the reference in lines 595 to 603 like this:

“To make the CSS value to be normally distributed, the rank-based inverse normal transformations (Rank-based INT) were applied to transforming the distribution of each OTU to make it appear more normally distributed as described in a previous study(Beasley et al., 2009). Rank-based INT entails creating a modified rank variable and then computing a new transformed value of the phenotype for the i th subject. A detailed description of the calculating formula is given below:

$$y_i^t = \Phi^{-1} \left(\frac{r^i - c}{N - 2c + 1} \right)$$

where r_i is the ordinary rank of the i th case among the N observations and Φ^{-1} denotes the standard normal quantile (or probit) function. The value of $c = 3/8$ is recommended by Blom (Blom, 1958).”

References:

Beasley, T. M., Erickson, S. & Allison, D. B. Rank-based inverse normal transformations are increasingly used, but are they merited? *Behavior genetics* 39, 580-595 (2009).

Blom, G. *Statistical estimates and transformed beta-variables*, Almqvist & Wiksell, (1958).

Line 494: Which type of media was used?

RESPONSE: We used the Nutrient Agar (NA) medium, which is a general-purpose medium for the cultivation of a wide variety of bacterial organisms. I add the manufacturer information (BD Company) in line 621 as follows:

“The supernatants were diluted from 10^{-1} to 10^{-7} and then 10^{-4} and 10^{-6} dilutions were distributed and cultivated in 96-well microtiter plates in Nutrient Agar (NA) media (BD Company) for 48-72h at 28 °C.”

Line 495: universal primers

RESPONSE: We added the primers information in the context as follows:

“The full length of 16S rRNA gene for each strain was amplified with bi-barcoded universe primers (27F 5'-GAGTTTGATCCTGGCTCAG-3'; 1492R 5'-TACCTTGTTACGACTT-3').”

Line 496: pooled together in equimolar ratios

RESPONSE: Yes, thanks. We revised the words “pooled together with the same amount” to “pooled together in equimolar ratios”.

As an aside: I would consider coming up with a naming convention for the marker OTUs. I understand it is easier to just mention their numbers and OTU ids, but it would benefit the reader if the associated taxonomy was easily accessible in the text or figures. For example, in Figure 2B and C, the bar plots could be ordered in an alphabetical order, or according to classes. This was already done in Figure 6B.

RESPONSE: Thanks for your helpful suggestion. We marked the strains with the front four letters of their taxonomies. We revised the marks in Figure 2A, B and C accordingly. We also introduced the responsive relationship between the marker OTUSs to the representative strains in the figure legend to make it more easily accessible to the reader.

Fig. S2, the Heritability estimates could be colored by phenotype category (e.g. growth versus yield)

RESPONSE: We have revised Figure S2 and marked the growth and yield box with red and green respectively.

Fig. S5 – a subset of the OTUs are highly heritable: Bacilalles and GP4??

RESPONSE: Yes, there are a higher number of highly heritable OTUs in order *Bacilalles* and *GP4* than others.

Fig. S7 – the Venn diagram labels are unclear. Please check the numbers on the right hand side, as they appear to be in error.

RESPONSE: Thanks. We have corrected the numbers marked on the right hand of the Venn diagram in Figure S10.

Reviewers' Comments:

Reviewer #1:

Remarks to the Author:

Thank you for your response. Plant's phenotypes are entirely different from the humans, so justification by quoting examples of human studies is not appropriate. Please see the following review -

Beilsmith K, Thoen MPM, Brachi B, Gloss AD, Khan MH, Bergelson J. Genome-wide association studies on the phyllosphere microbiome: Embracing complexity in host-microbe interactions. *Plant J.* 2019 Jan;97(1):164-181. doi: 10.1111/tpj.14170.

Phenotyping

The most important aspect of any GWAS pipeline is defining which traits will be mapped to the host genome. This in turn determines the type of raw phenotype data to collect. To help determine the type of microbiome sequencing to perform and the way in which data will be analyzed, I suggest focusing on three main questions.

1. Is the research question better addressed by associating host genes with microbial taxa or with the biological functions encoded in microbial genomes?
2. If the research focuses on taxa, what mode of host-microbe interactions is more likely: diffuse interactions between hosts and microbial communities or targeted interactions between hosts and specific microbes in the community?
3. How important is the environment in shaping host-microbe interactions, and do the hypotheses under consideration require explicit investigation of the impact of host genotype-by-environment interactions on microbial communities?

Reviewer #2:

Remarks to the Author:

I have accepted the author's revisions so that my questions and concerns were properly addressed. It is my intention to recommend acceptance of this manuscript to the editor.

Reviewer #4:

Remarks to the Author:

I was asked to assess the authors' response to the original 3rd reviewer, who was unable to complete the second round of review.

My overall impression of the article is quite positive. These will be valuable datasets to the research community. The finding that the inclusion of root microbiome data improves the power to explain phenotypic variation (above and beyond standard GWAS) is quite striking. The new Figure S3, produced in response to Rev. 3's comment, displays this nicely. The authors do a good job of acknowledging and dealing with the fact that OTUs themselves can be partially heritable, yet also influence the plant phenotype (either independently of plant genotype or interacting with it). The validation of marker OTUs using microbial isolation and re-inoculation is very valuable and impressive.

The response to reviewer 3 is thorough and generally sufficient, with a small exception:

"Line 104: what are tolerant genes?"

-- the response to this query was to list several examples of "drought stress-responsive" and "tolerant" genes, however, this does not quite solve the problem. A few more words are needed to provide an actual definition of "tolerant genes". E.g., are these genes whose expression remain consistent regardless of water availability?

Finally a few minor comments of my own:

-- Throughout the paper, please check the formatting of the names of bacterial taxa. Only species

and genus names should be italicized. Family level and above, should not be italicized.

-- Lines 67-68: This statement seems odd to me. Quantitative variation in traits, per se, has not been a barrier to genetic improvement as long as that variation is sufficiently heritable. Centuries of artificial selection have produced staggering changes in key quantitative traits of crop plants. Also, the citation #25 provided here might be wrong - the cited paper is about the role of cell death in root development.

-- Data availability: a search for the accession #CNP0001521 in CNGBdb does not find any results, can the authors confirm that this is not an error but just an embargo?

-- Data availability: A large amount of plant phenotype data, both from the field and from the inoculation experiments in plates and sterile soil, went into this paper. Where are these data available?

Dear Editor and Reviewers,

We greatly appreciate your constructive comments and suggestions. We have performed an additional analysis that has further validated and strengthened all of our major conclusions in the manuscript. Below we provide comments on each of their suggestions or concerns.

Reviewer #1 (Remarks to the Author):

Thank you for your response. Plant's phenotypes are entirely different from the humans, so justification by quoting examples of human studies is not appropriate. Please see the following review -

Beilsmith K, Thoen MPM, Brachi B, Gloss AD, Khan MH, Bergelson J. Genome-wide association studies on the phyllosphere microbiome: Embracing complexity in host-microbe interactions. *Plant J.* 2019 Jan;97(1):164-181. doi: 10.1111/tpj.14170.

Phenotyping

The most important aspect of any GWAS pipeline is defining which traits will be mapped to the host genome. This in turn determines the type of raw phenotype data to collect. To help determine the type of microbiome sequencing to perform and the way in which data will be analyzed, I suggest focusing on three main questions.

RESPONSE: Thanks for providing these excellent suggestions. We have responded to your specific queries in the below sections

1. Is the research question better addressed by associating host genes with microbial taxa or with the biological functions encoded in microbial genomes?

RESPONSE: Thanks for raising this point. In this study, we aim to unravel the molecular basis of host-microbe interactions by mGWAS and used this information to optimize plant growth. Our results uncover comprehensive plant genotype-microbiome networks that contribute to phenotype plasticity. Firstly, the transcripts of the associating host gene with marker strains varied between the control and treatment in sterilized soil experiments (Figure S13), indicating the interactions of host genes and marker strains. Secondly, marker stains Kita594 and Baci173 showed different effects on reference and allele genotype cultivars of foxtail millet in plate experiments, respectively. Significant effects of the interaction between the genotype and strain Kita594 and strain Baci173 on the shoot and root length were also detected by PERMANOVA, respectively (genotypes*Kita594: $R^2=13.048$, $P < 0.001$; genotypes*Baci173: $R^2=0.07$, $P < 0.001$, Table S13). These results supported that host genetic variation impacts the interactions between marker strains and host plants, finally affecting the plant phenotypes, consistent with our research question.

Of course, there are still many issues that need to be further explored, such as possible biological mechanisms underlying the associations. Multi-omics data such as host metabolome, transcriptome, microbial metatranscriptome *etc.*, and mutant experiments would help to answer these questions in the future study.

2. If the research focuses on taxa, what mode of host-microbe interactions is more likely: diffuse interactions between hosts and microbial communities or targeted interactions between hosts and specific microbes in the community?

RESPONSE: Thanks for your suggestions. We added the analysis to investigate the mode of host-microbe interactions (targeted or diffuse plant-microbe interactions) as the previous study described (Beilsmith et al., 2019). The topology characteristics (e.g. degree, betweenness centrality, closeness centrality and hub score) of the network from 1004 common OTUs were calculated using R packages igraph 1.2.11. We defined “Hub species” as OTU with high values of degree (> 400) and closeness centrality (> 0.5) in the network as previous study (Xiong et al., 2021). Thus 102 hub OTUs were identified in 1004 common OTUs, and these hub species were mainly affiliated within 22 microbial orders (hub taxa), such as Rhizobiales (20 hub OTUs), Actinomycetales (13), Solirubrobacterales (13), Gp6 (11), Gaiellales (7), Myxococcales (6), Xanthomonadales (4) and so on. The non-hub species belong to 50 orders and are mainly composed of Bacillales (165 OTUs), Actinomycetales (156), Sphingobacteriales (79), Burkholderiales (43), Xanthomonadales (41) and so on (Figure R1).

Figure R1 The taxonomic distributions of the hub and non-hub OTUs at the order level. The numbers of hub OTUs are colored yellow and the non-hub OTUs are colored blue.

Secondly, we identified the overlap of the hub and non-hub OTUs with the 838 SNP-associated OTUs. Among these, 90 hub OTUs and 748 non-hub OTUs had a significant association with the host genetic SNP loci, indicating host plant might interact by targeting with these hub microbes and diffusely interact with these non-hub microbes (Figure S10A, Table S10). We aggregate these SNP-associated hub OTUs (90 hub OTUs) and non-hub OTUs (748 OTUs) into 12 and 36 microbial orders, respectively. Comparative analysis showed that one order GP7 was only composed of SNP-associated hub OTUs, and 25 orders such as Sphingobacteriales, Bacillales, Ohtaekwangia, Sphingomonadales and Acidimicrobiales were only composed of SNP-associated non-hub OTUs

and 11 orders were composed of both hub and non-hub OTUs (Figure S10B). These data indicated that the foxtail millet target interacted with several hub species and diffusely interacted with most of the microbial species.

Figure S10 Characterization of the genotype-associated hub and non-hub OTUs in the rhizoplane microbiota. A. Venn diagram depicting the numbers of SNP-associated hub OTUs and non-hub OTUs. B Venn diagram depicting the shared and specific taxa between the SNP-associated hub and non-hub tax at the order level.

To decipher the potential mechanism of the targeted interactions between the foxtail millet and hub microbes, the candidate host genes around the SNP loci that associated with the hub OTUs were extracted and the functions were mainly involved in plant growth and development process, plant immunity system, transcription factor, nutrient uptake, metabolites, abiotic stress response and others (Table S10 and Figure S11A). Specifically, the immune gene FLS2 is widely associated with 10 bacterial orders covering 25 hub OTUs (24.5% of all hubs), such as GP6 (10 hub OTUs), Rhizobiales (3), Burkholderiales (3) and Actinomycetales (2), indicating the key function of FLS2 in regulating root microbial composition. It is well known that FLS2 functioned as the bona fide pattern-recognition receptor for flagellin involved in plant and microbe interaction (Chinchilla et al., 2006), which could regulate the microbiome assembly in planta (Chen et al., 2020). The transcription factor bHLH35 is mainly associated with hub taxa Burkholderiales (2 hub OTUs), Gp3 (1), Gp6 (9) and Rhizobiales (1). bHLH35 proteins are transcription factors induced by effector-triggered immunity (ETI), and involved in tolerance to abiotic stresses (Jiang et al., 2019). In addition, the host gene WAK2 is mainly associated with the Acidobacteria GP6 (5 hub OTUs), GP7 (2), GP4 (1), GP3 (1). WAK2 has been reported with functions in activating a stress response that includes an increased ROS accumulation and the up-regulation of numerous genes involved in pathogen resistance, wounding, and cell wall biogenesis (Kohorn et al., 2009; Kohorn et al., 2012), which might impact the abundance of Acidobacteria.

Thirdly, the host genes associated with the non-hub OTUs were also examined. Obviously, the gene associated with the non-hub taxa varied among the different taxa except for widely regulated genes FLS2 and bHLH35 (Figure S11A and B). For example, the genes associated with taxa

Actinomycetales, Bacillales, Spingobacteriales, Burkholderiales and Rhizobiales are totally different, showing a taxa-dependent regulation model (Figure S11B and C).

Fig S11 The association networks of host genes with hub taxa (A) and non-hub taxa (B). C. Venn diagram depicting the differences of the associating host genes among the different non-hub taxa.

Generally, the foxtail millet employed two modes to structure the root microbiome: targeted interacted with several hub species and diffusely interacted with most of the microbial species. Although the host immune genes FLS2 and transcription factor bHLH35 are widely associated with the hub and non-hub microbes, the host plant still employed other different genes to interact with different taxa, suggesting a taxa-dependent regulation model.

We have added the analysis results in the manuscript and updated the content as follows:

“As plants primarily influence their microbiomes through targeted interactions with key taxonomic groups or diffuse interactions with entire communities⁵⁵. To further investigate the mode of host-microbe interactions, the hub microbial taxa and non-hub microbial taxa and their associated genes were identified. Firstly, we defined hub taxa as OTU with high values of degree (> 400) and closeness centrality (> 0.5) in the network as described in a previous study⁵⁶, resulting in 102 hub OTUs. We identified that 90 hub OTUs and 748 non-hub OTUs had significant associations with the host genetic SNP loci (Figure S10A, Table S10), indicating host plant might targeted interact with these hub microbes and diffusely interact with these non-hub microbes. We aggregated these SNP-associated hub OTUs (90 hub OTUs) and non-hub OTUs (748 OTUs) into 12 and 36 microbial orders, respectively. Comparative analysis showed that one order GP7 was only composed of SNP-associated hub OTUs, and 25 orders such as Spingobacteriales, Bacillales, Ohtaekwangia, Spingomonadales and Acidimicrobiales were only composed of SNP-associated non-hub OTUs, and 11 orders were composed of both SNP-associated hub and non-hub OTUs (Figure S10B). These

data indicated that the foxtail millet employed two modes to structure the root microbiome: targeted interaction with several hub microbes and diffused interaction with most of the microbes. To decipher the potential mechanism of the interaction between plant and microbe, the candidate host genes around the SNP loci associated with the hub and non-hub OTUs were extracted separately. The networks showed that the host immune genes FLS2 and transcription factor bHLH35 are widely associated with the hub and non-hub taxa (Figure S11A and B). However, the host plant still employed different genes to interact with different taxa (Figure S11C), suggesting a taxa-dependent regulation model.”

References:

- Beilsmith, K., Thoen, M.P., Brachi, B., Gloss, A.D., Khan, M.H., and Bergelson, J. (2019). Genome-wide association studies on the phyllosphere microbiome: embracing complexity in host–microbe interactions. *The Plant Journal* 97, 164-181.
- Chen, T., Nomura, K., Wang, X., Sohrobi, R., Xu, J., Yao, L., Paasch, B.C., Ma, L., Kremer, J., and Cheng, Y. (2020). A plant genetic network for preventing dysbiosis in the phyllosphere. *Nature* 580, 653-657.
- Chinchilla, D., Bauer, Z., Regenass, M., Boller, T., and Felix, G. (2006). The Arabidopsis receptor kinase FLS2 binds flg22 and determines the specificity of flagellin perception. *The Plant Cell* 18, 465-476.
- Jiang, L., Tian, X., Li, S., Fu, Y., Xu, J., and Wang, G. (2019). The AabHLH35 transcription factor identified from *Anthurium andraeanum* is involved in cold and drought tolerance. *Plants* 8, 216.
- Kohorn, B.D., Johansen, S., Shishido, A., Todorova, T., Martinez, R., Defeo, E., and Obregon, P. (2009). Pectin activation of MAP kinase and gene expression is WAK2 dependent. *The Plant Journal* 60, 974-982.
- Kohorn, B.D., Kohorn, S.L., Todorova, T., Baptiste, G., Stansky, K., and McCullough, M. (2012). A dominant allele of Arabidopsis pectin-binding wall-associated kinase induces a stress response suppressed by MPK6 but not MPK3 mutations. *Molecular plant* 5, 841-851.
- LI, Z.-j., JIA, G.-q., LI, X.-y., LI, Y.-c., Hui, Z., Sha, T., ZHANG, S., LI, Y.-d., SHANG, Z.-l., and DIAO, X.-m. (2021). Identification of blast-resistance loci through genome-wide association analysis in foxtail millet (*Setaria italica* (L.) Beauv.). *Journal of Integrative Agriculture* 20, 2056-2064.
- Xiong, C., He, J.Z., Singh, B.K., Zhu, Y.G., Wang, J.T., Li, P.P., Zhang, Q.B., Han, L.L., Shen, J.P., and Ge, A.H. (2021). Rare taxa maintain the stability of crop mycobiomes and ecosystem functions. *Environmental Microbiology* 23, 1907-1924.

3. How important is the environment in shaping host-microbe interactions, and do the hypotheses under consideration require investigation of the impact of host genotype-by-environment interactions on microbial communities?

RESPONSE: Thanks for raising this point. We didn't consider the impacts of host genotype-by-environment interactions on microbial communities in this study. Thus, we collected the root microbiome samples from foxtail millet cultivars in a single environment, and changes in effects (genotype-by-environment interactions) are consistent across the environments. Using the environment as a variable can be used to find polymorphisms that have robust effects on the

microbiome across environments, and a clear resolution of these questions will require a comprehensive future study.

Together with validation of the microbial-mediated growth effects on foxtail millet with marker strains isolated from the field in our manuscript. We hope that you would kindly agree with us that all major conclusions of this foundational paper are sufficiently supported by the data presented, which set an important starting point for addressing many exciting questions by my lab and other research groups in the coming years.

Reviewer #2 (Remarks to the Author):

I have accepted the author's revisions so that my questions and concerns were properly addressed. It is my intention to recommend acceptance of this manuscript to the editor.

RESPONSE: Thanks for endorsing our manuscript. Your comments were highly insightful and enabled us to greatly improve the quality of our manuscript.

Reviewer #4 (Remarks to the Author):

I was asked to assess the authors' response to the original 3rd reviewer, who was unable to complete the second round of review.

My overall impression of the article is quite positive. These will be valuable datasets to the research community. The finding that the inclusion of root microbiome data improves the power to explain phenotypic variation (above and beyond standard GWAS) is quite striking. The new Figure S3, produced in response to Rev. 3's comment, displays this nicely. The authors do a good job of acknowledging and dealing with the fact that OTUs themselves can be partially heritable, yet also influence the plant phenotype (either independently of plant genotype or interacting with it). The validation of marker OTUs using microbial isolation and re-inoculation is very valuable and impressive.

The response to reviewer 3 is thorough and generally sufficient, with a small exception:

"Line 104: what are tolerant genes?"

-- the response to this query was to list several examples of "drought stress-responsive" and "tolerant" genes, however, this does not quite solve the problem. A few more words are needed to provide an actual definition of "tolerant genes". E.g., are these genes whose expression remain consistent regardless of water availability?

RESPONSE: Thanks for recommending our manuscript and raising this important point. The drought tolerant genes represent the genes related to signaling and gene transcription, are frequently upregulated in drought-stressed seedlings and could promote stomatal closure, reduced water loss and enhanced drought tolerance in plants.

We have added a definition in the content and revised the sentence as follows:

“Secondly, numerous drought stress-responsive (*PP2C*, *ARR12*, *NPF1.2*, *NPF4.6*, *WDR26*, *Plastocyanin-like protein*, *CPK2a*, *PIP5K1*) and tolerant genes (*APX*, *DTX12*, *bHLH3*, *Thioredoxin fold domain containing protein*, *SAPK9*, *Ca²⁺-transporting ATPase*, *InsP3*, *E3 ubiquitin-protein*)

ligase MIELI) whose expression are frequently upregulated and contribute to drought resistance in drought-stressed seedlings, were found to be strongly associated with the growth and yield traits.”

Finally a few minor comments of my own:

-- Throughout the paper, please check the formatting of the names of bacterial taxa. Only species and genus names should be italicized. Family level and above, should not be italicized.

RESPONSE: Thanks. We have checked the manuscript thoroughly and revised the formatting of the names of bacterial taxa according to your suggestions.

-- Lines 67-68: This statement seems odd to me. Quantitative variation in traits, per se, has not been a barrier to genetic improvement as long as that variation is sufficiently heritable. Centuries of artificial selection have produced staggering changes in key quantitative traits of crop plants. Also, the citation #25 provided here might be wrong - the cited paper is about the role of cell death in root development.

RESPONSE: Thanks for your comments. We have revised the sentence and corrected the reference as follows:

“Virtually all yield component traits and most agronomic traits of foxtail millet are quantitative inheritance (Fang et al., 2016), GWAS has revealed key loci for early and late flowering times and blast-resistance in foxtail millet (Jia et al., 2013; LI et al., 2021), however, the loci associated with plant growth or yield are still not known.”

Reference:

Fang, X. et al. A high density genetic map and QTL for agronomic and yield traits in Foxtail millet [*Setaria italica* (L.) P. Beauv.]. *BMC genomics* 17, 1-12 (2016).

Jia, G. et al. A haplotype map of genomic variations and genome-wide association studies of agronomic traits in foxtail millet (*Setaria italica*). *Nature genetics* 45, 957-961 (2013).

LI, Z.-j. et al. Identification of blast-resistance loci through genome-wide association analysis in foxtail millet (*Setaria italica* (L.) Beauv.). *Journal of Integrative Agriculture* 20, 2056-2064 (2021).

-- Data availability: a search for the accession #CNP0001521 in CNGBdb does not find any results, can the authors confirm that this is not an error but just an embargo?

RESPONSE: We have applied for a temporarily reviewer link and you can access the data via this link (http://db.cngb.org/cnsa/project/CNP0001521_e229a6d2/reviewlink/), and we assure you that all of the sequencing data would be open to the public after the acceptance of the manuscript.

-- Data availability: A large amount of plant phenotype data, both from the field and from the inoculation experiments in plates and sterile soil, went into this paper. Where are these data available?

RESPONSE: We have attached the experiment data with the analysis script together and uploaded all files to the Github repository. The experiment data for figures 3B-C and figure 6C-H could be found in the corresponding files under the public link “<https://github.com/wyy2017/Foxtail-millet-data-analysis>”.

Reviewers' Comments:

Reviewer #1:

Remarks to the Author:

The study generated good data and interpretations which is appreciated. Some comments are also fulfilled by conducting additional analysis. However, ignorance of environmental effects and genotype x environment interactions are major lacunae of the present GWAS study, which is essential while studying complex traits.

Dear Reviewers,

We greatly appreciate your constructive comments and suggestions. We have responded to your specific queries in the below sections.

REVIEWER COMMENTS

Reviewer #1 (Remarks to the Author):

The study generated good data and interpretations which is appreciated. Some comments are also fulfilled by conducting additional analysis. However, ignorance of environmental effects and genotype x environment interactions are major lacunae of the present GWAS study, which is essential while studying complex traits.

RESPONSE: Thanks for your comments. We greatly appreciate your constructive comments and suggestions to guide revision of previous manuscript. We have toned down the claims on the results of mGWAS and highlighted the new text in the manuscript to address your concerns (Lines 457 to 468).

“The differences in the field environments may cause distinct loci or host processes to shape the root microbiome. Independent GWAS in multiple environments or single environment could have both the same and different association features, but with different emphasis. In this study, numerous potential genes associated with the root microbiome were identified by mGWAS based on 827 foxtail millet cultivars in a single environment, and changes in effects (genotype-by-environment interactions) are consistent across the environments. Among these, the genes such as *FLS2*, *WAK2* and GDSL-type lipase have been reported to function in structuring the plant microbiome assembly or mediate pathogen and stress responses^{11,52,59}. Of course, numerous genes governing the microbial effect in this study were excavated for the first time and yet to be proved by further experiments. In future, independent GWAS performed in multiple environments will enable the dissection of shared genetic loci which likely affect the root microbial communities.”